# Distinct mechanisms underlie $H_2O_2$ sensing in *C. elegans* head and tail

Sophie Quintin[1,2,3,4]*, Théo Aspert[1,2,3,4], Tao Ye[1,2,3,4], Gilles Charvin[1,2,3,4]

**1** Department of Developmental Biology and Stem Cells, Institut de Génétique et de Biologie Moléculaire et Cellulaire, Illkirch, France, **2** Centre National de la Recherche Scientifique, UMR7104, Illkirch, France, **3** Institut National de la Santé et de la Recherche Médicale, U964, Illkirch, France, **4** Université de Strasbourg, Illkirch, France

* quintin@igbmc.fr

**Data Availability Statement:** All relevant data are within the paper and its Supporting Information files.

**Funding:** 'This work was funded by the Agence Nationale de la Recherche (grants ANR-10-

## Abstract

Environmental oxidative stress threatens cellular integrity and should therefore be avoided by living organisms. Yet, relatively little is known about environmental oxidative stress perception. Here, using microfluidics, we showed that like I2 pharyngeal neurons, the tail phasmid PHA neurons function as oxidative stress sensing neurons in *C. elegans*, but display different responses to $H_2O_2$ and light. We uncovered that different but related receptors, GUR-3 and LITE-1, mediate $H_2O_2$ signaling in I2 and PHA neurons. Still, the peroxiredoxin PRDX-2 is essential for both, and might promote $H_2O_2$-mediated receptor activation. Our work demonstrates that *C. elegans* can sense a broad range of oxidative stressors using partially distinct $H_2O_2$ signaling pathways in head and tail sensillae, and paves the way for further understanding of how the integration of these inputs translates into the appropriate behavior.

## Introduction

Reactive oxygen species (ROS) are well-known to exert a dual effect, promoting aging and pathological conditions on the one hand and increasing organism resistance and longevity on the other hand [1]. Neurons are easily exposed to ROS, and many neurodegenerative diseases have been associated with oxidative stress [2]. However, exposure of *C. elegans* nematodes to a mild oxidative stress has been reported to be beneficial for neuron sensory function: micromolar doses of the ROS-inducing agent paraquat or of hydrogen peroxide ($H_2O_2$) improve the sensitivity in ASH polymodal neurons [3, 4], whereas millimolar doses of $H_2O_2$ reduce neuron response [3], likely inducing oxidative stress. Therefore, it is essential for nematodes to detect a broad range of $H_2O_2$ concentrations to preserve their cellular integrity.

Although the cellular response to oxidative stress has been extensively characterized (reviewed in [5]), little is known on how oxidants are actually perceived and which underlying molecular pathways are involved. A recent study has revealed the behavioral mechanisms which allow *C. elegans* to find a niche providing both food and protection from $H_2O_2$ [6]. In addition, light and $H_2O_2$ sensing were shown to be tightly connected in yeast [7] and in nematodes [8]. While it has been demonstrated that light is converted into an $H_2O_2$ signal in yeast

LABX0030-INRT to Gilles Charvin and ANR-10-IDEX-0002; ANR 20-SFRI-0012; ANR-17-EURE0023 to the Interdisciplinary Thematic Institute IMCBio of the University of Strasbourg, CNRS and Inserm, part of the 2021-2028 Investments for the Future Program.

**Competing interests:** The authors declare no competing interest.

[7], this question remains unanswered in nematodes. Notably, *C. elegans* can detect both H$_2$O$_2$ and light via the I2 pharyngeal neurons and responds to these stimuli by inhibition of feeding or by an avoidance behavior [8]. Initially described as interneurons, the I2 neurons proved to be primary sensory neurons [9] that are highly specialized in oxidative stress sensing [8], but were also shown to respond moderately to salt or odor [10]. However, although detection of a large spectrum of H$_2$O$_2$ is critical for nematodes, the range of H$_2$O$_2$ concentrations detected by I2 neurons has not been investigated, and the molecular mechanisms involved in H$_2$O$_2$ signaling are not well defined.

In addition, the nematode also possesses tail sensory neurons specialized in chemo-repulsion, called PHA/PHB (or phasmids, reviewed in [11]). In contrast to I2 neurons, these tail neurons can respond to many noxious stimuli [12] and trigger avoidance [13], but whether PHA neurons can sense H$_2$O$_2$ or light remains an open question.

H$_2$O$_2$ sensing in I2 neurons requires the function of the peroxiredoxin PRDX-2 [8], a highly conserved antioxidant enzyme whose role remains unclear. Peroxiredoxins (Prxs) belong to a family of thiol peroxidases which can reduce H$_2$O$_2$ in cells following the oxidation of one or two cysteines in their catalytic domain, and they are highly abundant from yeast [14, 15] to human cells, in which they represent approximately 1% of the total dry cellular mass [16, 17]. Oxidised Prxs are recycled in the reduced, active form, by thioredoxin [18]. At high H$_2$O$_2$ concentrations, Prxs become hyperoxidized, a form that has been shown to function as a molecular chaperone [19, 20]. Similarly, thioredoxins have been shown to have redox-independent function [21, 22]. As pivotal antioxidants, Prxs dysfunction has been associated with several pathologies [23], including cancer [24]. In budding yeast, the peroxiredoxin Tsa1 has a major role in maintaining the redox balance, and is massively induced upon oxidative stress as a direct target of the H$_2$O$_2$-sensing transcription factor Yap1 [25, 26]. In *C. elegans*, among three genes encoding peroxiredoxins, PRDX-2 is the only one whose depletion induces a phenotype and is considered as the major peroxiredoxin [27, 28]. PRDX-2 is expressed in many cell types including neurons, gut [27–29], muscle and epithelial cells [8]. Although a global induction of *prdx-2* mRNA expression has been reported upon treatment with the strong tBOOH oxidant [28], cells in which PRDX-2 expression is induced remain to be identified. Likewise, the question of a tissue-specific regulation of PRDX-2 by SKN-1, the closest ortholog of Yap1, has not been addressed.

Importantly, beyond their peroxidase activity, Prxs have long been proposed to act as intracellular H$_2$O$_2$ sensors, which influence cellular signaling [30–32]. For example, the p38/MAPK signaling pathway, which controls adaptive mechanisms and/or cell fate decisions, is activated by H$_2$O$_2$ through Prxs in an evolutionary conserved manner [33, 34]. Like in mammalian or in drosophila cells, several studies in *C. elegans* indicate that PRDX-2 would relay H$_2$O$_2$ signaling, activating the downstream p38/PMK-1 pathway [3, 35, 36]. Notably, low doses of H$_2$O$_2$ potentiate the sensory response of ASH sensory neurons to glycerol through activation of the PRDX-2/p38/PMK-1 cascade [3]. Yet, whether this cascade is at play in I2 neurons to influence H$_2$O$_2$ perception has not been analyzed.

Here, we undertook a subcellular analysis of the peroxiredoxin PRDX-2 in *C. elegans*, focusing on its requirement in neuronal H$_2$O$_2$ sensing. Using a CRISPR knock-in line, we showed that PRDX-2 is present in many cells, among which several pairs of neurons: I2s in the head, PHAs in the tail and CANs in the body. Interestingly, upon an H$_2$O$_2$ challenge, an upregulation of PRDX-2 is observed only in the anterior gut and in the excretory pore, but not in neurons, suggesting that PRDX-2 might fulfill different functions, depending on the cell type where it is expressed. Using a microfluidic-based approach and real-time calcium imaging, we show that PHA neurons also respond to H$_2$O$_2$, with an even higher sensitivity than I2 neurons. Although H$_2$O$_2$ perception depends on *prdx-2* function in both pairs of neurons, we

uncovered that it relies on different gustatory receptors and downstream transducers: while dispensable in I2 neurons, the p38/MAPK kinase pathway contributes to the hypersensitivity of PHA neurons to $H_2O_2$. Interestingly, we uncovered that $H_2O_2$ sensing requires the same receptors as light sensing, and that PHA neurons respond to light—establishing a parallel between PHA and photosensory ASH neurons. Based on our work and on previous studies, we propose a molecular model of how $H_2O_2$ could trigger neuronal activation in I2 and PHA through a peroxiredoxin-mediated redox relay. Taken together, our data suggest that *C. elegans* can sense a broad range of oxidative stress using partially distinct $H_2O_2$ signaling pathways acting in head and tail sensillae.

## Results

### Expression pattern of PRDX-2::GFP and its variation upon $H_2O_2$ treatment

In budding yeast, the peroxiredoxin Tsa1 is massively induced upon $H_2O_2$ treatment [25, 26]. To gain insight into the tissue-specific expression of PRDX-2 upon oxidative stress, we first used the PRDX-2 reporter line generated by [29]. However, the level of expression of this strain varies a lot, and transgenics show many fluorescent aggregates ([29], S1 Fig), likely associated with transgene overexpression. Consistently, the strain displays a much stronger resistance to oxidative stress than wild-type animals (S1 Fig), suggesting that overexpressed PRDX-2 construct induces a higher $H_2O_2$ scavenging capacity in transgenics. Furthermore, it was impossible to identify PRDX-2-expressing cells in this strain, preventing the use of this strain in our study. For these reasons, we created a GFP knock-in line of PRDX-2, using the CRISPR-Cas9 technique [37]. A C-terminus GFP fusion targeting all PRDX-2 isoforms, comprising a linker, was engineered and inserted at the *prdx-2* locus (Fig 1A). Three independent knock-in lines were obtained, sharing an identical expression pattern (Figs 1B and S1).

In the PRDX-2::GFP knock-in line, we detected a broad expression of PRDX-2 in various cell types, including the proximal and distal gut, muscles (body wall, vulval and pharyngeal), epithelial cells (vulva, nose tip, hypodermis) and I2 neurons (Fig 1B), as previously reported [8, 27, 28]. In addition, we observed for the first time PRDX-2 expression in the excretory pore cell (EPC), and in two other pairs of neurons; the tail phasmids (PHA/PHB, whose identity was ascertained by DiI staing, S1 Fig) and the excretory canal-associated neurons (CANs), located close to the vulva (Fig 1B). Consistent with this, a high number of *prdx-2* transcripts was detected in CANs and in the EPC [38]. Therefore, we hypothesized that the knock-in line faithfully reflects the endogenous PRDX-2 expression, and characterized it further.

### PRDX-2::GFP is induced in the anterior gut upon $H_2O_2$ treatment, but not in I2 neurons

We noticed that many PRDX-2-expressing cells are directly in contact with the environment, such as the EPC, the tip of the nose, the vulva, and neurons, which all possess terminations exposed outside. This pattern is strikingly reminiscent of that detected in animals carrying the *HyPer* $H_2O_2$ biosensor after $H_2O_2$ exposure [39]. Thus, PRDX-2 is expressed in cells in which environmental $H_2O_2$ penetrates more easily, suggesting a protective role of PRDX-2 in these cells as a peroxidase. Therefore, in the following, we wondered whether all PRDX-2-expressing cells respond similarly to an $H_2O_2$-induced oxidative stress.

To determine whether PRDX-2 expression changes upon oxidative stress, we exposed animals to different doses of $H_2O_2$, and PRDX-2::GFP expression was quantified in different cells after spinning-disc confocal acquisitions. We selected $H_2O_2$ concentrations inducing different

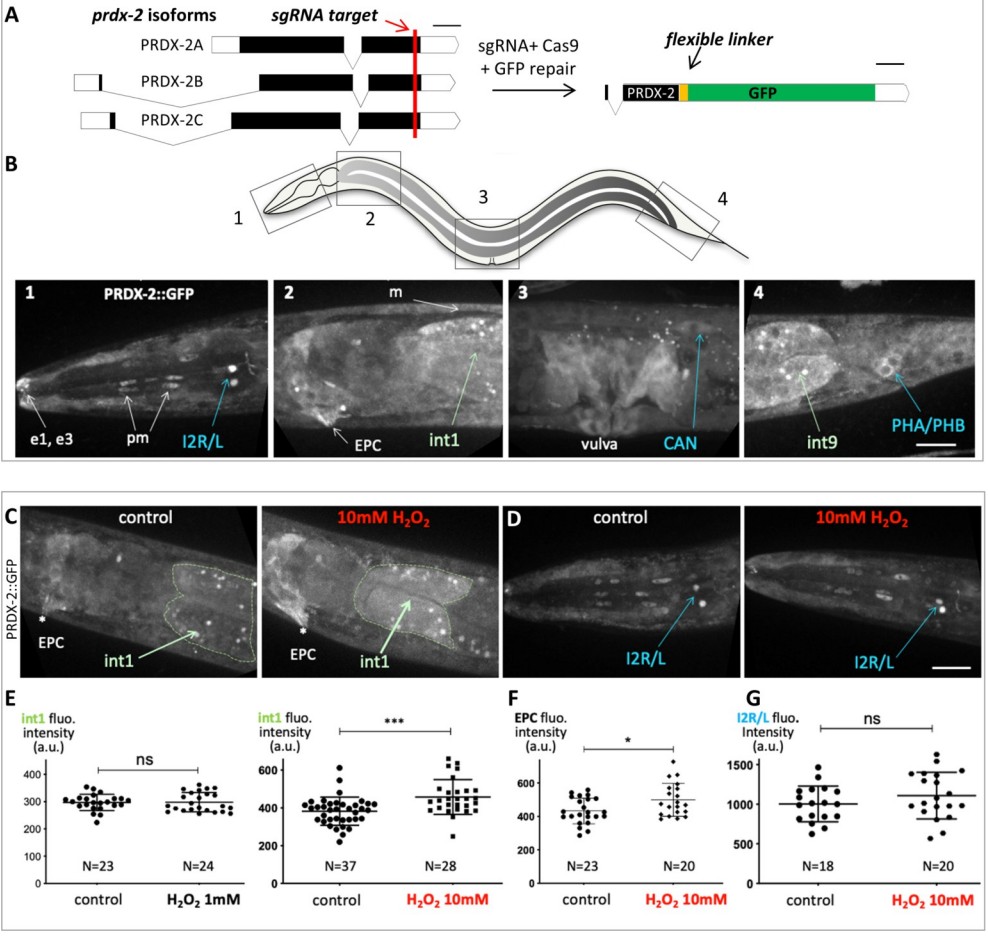

**Fig 1. PRDX-2::GFP knock-in line expression pattern and its variation upon H₂O₂ treatment.** (A) Sketch depicting the PRDX-2::GFP knock-in strategy using CRISPR Cas9-mediated genome editing. The sgRNA target sequence (shown in red) was chosen upstream PRDX-2 STOP codon, to tag all *prdx-2* isoforms. Black boxes indicate exons, white boxes untranslated regions. Bar, 100 bases pairs. The resulting last exon of the PRDX-2::GFP fusion protein is shown (right); note the insertion of a flexible linker between PRDX-2 and the GFP, to promote correct folding. After injection, three independent knock-in lines were recovered, sharing the same expression pattern. (B) Spinning-disc confocal projections of a representative PRDX-2::GFP knock-in animal, in 4 body regions (boxed in the worm sketch on top). PRDX-2::GFP expression is observed in the tip of the nose (e1, e3 epithelial cells), in pharyngeal muscle cells (pm), in body wall muscles (m), in the excretory pore cell (EPC), in proximal and distal gut cells (int1 and int9), and in several neuron pairs (I2, CAN, PHA/PHB). Note the different level of PRDX-2::GFP in I2 left and right neurons in panel B1. (C-G) An acute oxidative stress triggers an upregulation of PRDX-2 in the anterior gut, but not in neurons. (C-D) Spinning-disc confocal projections of control or H₂O₂-treated animals in the anterior gut (C) or in the head (D). (E-G) Quantification of fluorescence intensity in controls and in H₂O₂-treated animals in the int1 cell (E), in the EPC (F), and I2 neurons (G). Bars indicate mean and SD. ns, not significant, p>0.05; *p<0.05; ***p<0.001 (t test and Mann Whitney test). Scale bar, 20μm.

physiological responses [8]. A higher PRDX-2 expression was detected in the anterior gut two hours after a 10mM H₂O₂ treatment, but not after a 1mM H₂O₂ treatment (Fig 1C and 1E). The fluorescence was specific to PRDX-2::GFP as H₂O₂-treated controls did not show a higher gut autofluorescence (S2 Fig). Similarly, PRDX-2 expression was only induced in the EPC at 10mM H₂O₂ (Figs 1C and 1F and S2). Thus, our data indicate a dose-dependent PRDX-2 induction—as it occurs at 10mM but not at 1mM. The origin of this difference could arise from the animal behavioral response: at 1mM H₂O₂ the pharyngeal pumping is strongly inhibited to prevent ingestion [8], thereby exposure of gut to H₂O₂. This could explain both the

absence of PRDX-2 induction in the gut we report, and the unchanged level of *prdx-2* mRNA reported [28], after a 1mM H$_2$O$_2$ treatment. In contrast, at 10mM H$_2$O$_2$, the nematode should retract back (avoidance response, [8]); but here, as trapped animals cannot escape, they likely swallow a certain amount of H$_2$O$_2$, exposing the foregut to a severe oxidative stress which could trigger PRDX-2 induction. A 10mM H$_2$O$_2$ treatment of 30min has been shown to induce hyperoxidation of over 50% total PRDX-2 in wild-type lysed worms [40] and this form persists after a 4h recovery period [27]. Although the induction we observed is only based on expression level and not on protein activity, we suggest that PRDX-2 could still scavenge H$_2$O$_2$ in the EPC and in the foregut, consistent with the reported protective role of intestinal PRDX-2 [27].

Among PRDX-2-expressing neurons, the I2 pair shows the highest expression and possesses terminations exposed to the outside [9]. Therefore, we focused on I2 neurons to investigate whether the neuronal level of PRDX-2 is affected upon H$_2$O$_2$ treatment. However, at both concentrations tested (1 and 10mM), the level of PRDX-2 in I2 neurons constantly remained unchanged after H$_2$O$_2$ treatment (Figs 1D and 1G and S2).

Thus, we observed an upregulation of PRDX-2::GFP in the anterior gut and in the EPC following an acute oxidative stress, but not in neurons. As Prxs belong to a homeostatic system, an upregulation was shown to be associated with a detoxification function in yeast [26]. By analogy, PRDX-2 might fulfill a peroxidase function in the foregut and in the EPC to protect the animal against environmental aggressions. In contrast, in I2 neurons, PRDX-2 would instead act in H$_2$O$_2$ sensing and/or signaling, as proposed [8, 28]. We conclude that the responses of PRDX-2 to oxidative stress are cellular context-dependent.

## SKN-1 controls expression of PRDX-2 in the gut, but not in neurons

This prompted us to test whether PRDX-2 expression could rely on different subcellular regulations. By analogy with yeast, we wondered whether PRDX-2 would be controlled by the Yap1 nematode orthologue SKN-1 [5] in cells where PRDX-2 is induced upon oxidative stress, and asked whether such regulation occurs in other cells. We sought to test this hypothesis by inactivating the function of SKN-1 in the PRDX-2::GFP knock-in line. We first used the *skn-1 (zj15)* allele, which specifically inactivates gut-specific isoforms, leaving neuronal isoforms unaffected [41]. In this mutant, the basal expression level of PRDX-2 is reduced in the foregut, but an induction persists after a 10mM H$_2$O$_2$ treatment (Fig 2C), suggesting that the neuronal isoforms could mediate this effect. Consistent with this, the RNAi-mediated knock-down of all *skn-1* isoforms also triggered a reduction of PRDX-2::GFP signal in the anterior gut, and an absence of induction upon H$_2$O$_2$ treatment (Fig 2A, 2B and 2D). This result indicates that SKN-1 activity is essential to regulate PRDX-2 expression in the anterior gut, both at the basal level and under oxidative stress. In agreement with this, SKN-1 was found to bind to the *prdx-2* promoter in chromatin immunoprecipitation experiments coupled with high-throughput DNA sequencing (ChIP-seq) [42] (S3 Fig).

In contrast, in I2 neurons, no change in the PRDX-2 level was detected at the basal level or after H$_2$O$_2$ treatment compared to controls, neither in *skn-1(zj15)* mutants nor in *skn-1(RNAi)* animals (S2 Fig). With the limitation that RNAi efficiency may be lower in neurons [43], we suggest that additional transcription factor(s) might regulate *prdx-2* expression in these neurons. Consistently, several transcription factors were reported to bind to the *prdx-2* promoter by ChIP-seq [42] (S3 Fig).

We conclude that *skn-1* accomplishes a cell autonomous regulation of *prdx-2* in the intestine, but that this regulation may not occur in neurons. Taken together, our data suggest that

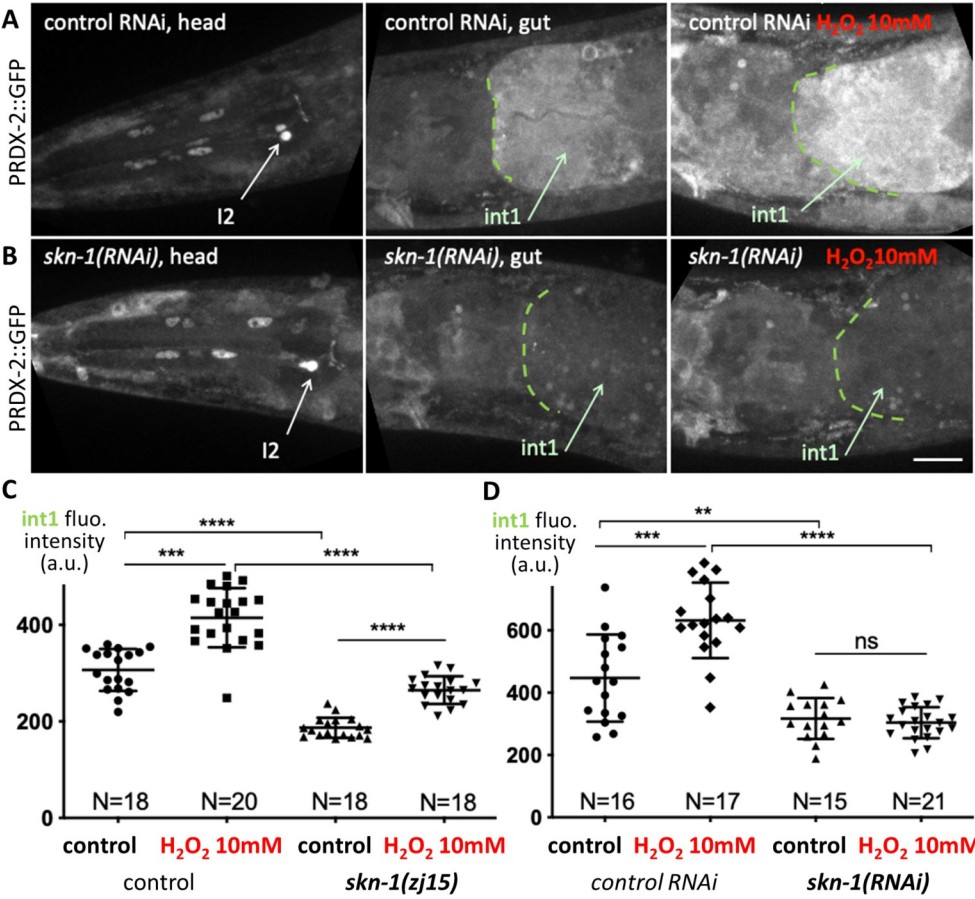

**Fig 2. SKN-1 function is required for PRDX-2 expression in the gut.** (A-B) Spinning-disc confocal projections of head and anterior gut of PRDX-2::GFP knock-in animals, in control RNAi (A) and in *skn-1(RNAi)* (B) animals. The right panel shows the foregut of a 10mM $H_2O_2$-treated animal in both genotypes. Note the low level of int1 PRDX-2:: GFP fluorescence in *skn-1(RNAi)* animal. (C-D) Quantification of the int1 cell fluorescence intensity in control and after a 10mM $H_2O_2$-treatment, in *skn-1(zj15)* mutants (C) and in *skn-1(RNAi)* animals (D). Bars indicate mean and SD. ns, not significant, $p>0.05$; **$p<0.01$; ***$p<0.001$; ***$p<0.0001$ (t test, ANOVA). Scale bar, 20µm.

PRDX-2 might exert a different function depending on the cell type: $H_2O_2$ detoxification in the foregut, triggered by SKN-1, *vs.* $H_2O_2$ perception or signaling in I2 neurons.

## PHA neurons respond to $H_2O_2$ in a prdx-2-dependent manner

As PRDX-2 is expressed in other neurons than I2s, we asked whether PRDX-2 endows these neurons with oxidative stress sensing properties. We focused on PHA tail neurons, as they belong to the phasmid sensory sensilla and respond to many noxious stimuli [12]. We monitored PHA neuron activation by imaging calcium using the GCaMP3 fluorescent sensor [44] expressed under the *flp-15* promoter, specific to I2 and PHA neurons [45].

To image calcium fluxes in neurons, L4 animals were trapped in microfluidic chambers and exposed to $H_2O_2$ (Figs 3A and S4), and their response was followed in 4D spinning-disc ultrafast acquisitions. This experimental setup, combined with semi-automated image analyses to quantify the mean fluorescence in neurons over time (Supplementary Information), confirmed the activation of I2 neurons upon a 1mM $H_2O_2$ treatment (Fig 3B and 3D and S1 Movie and S5 Fig), similar to that observed upon exposure to $H_2O_2$ vapor (*ie.* 8.82M, [8]). As

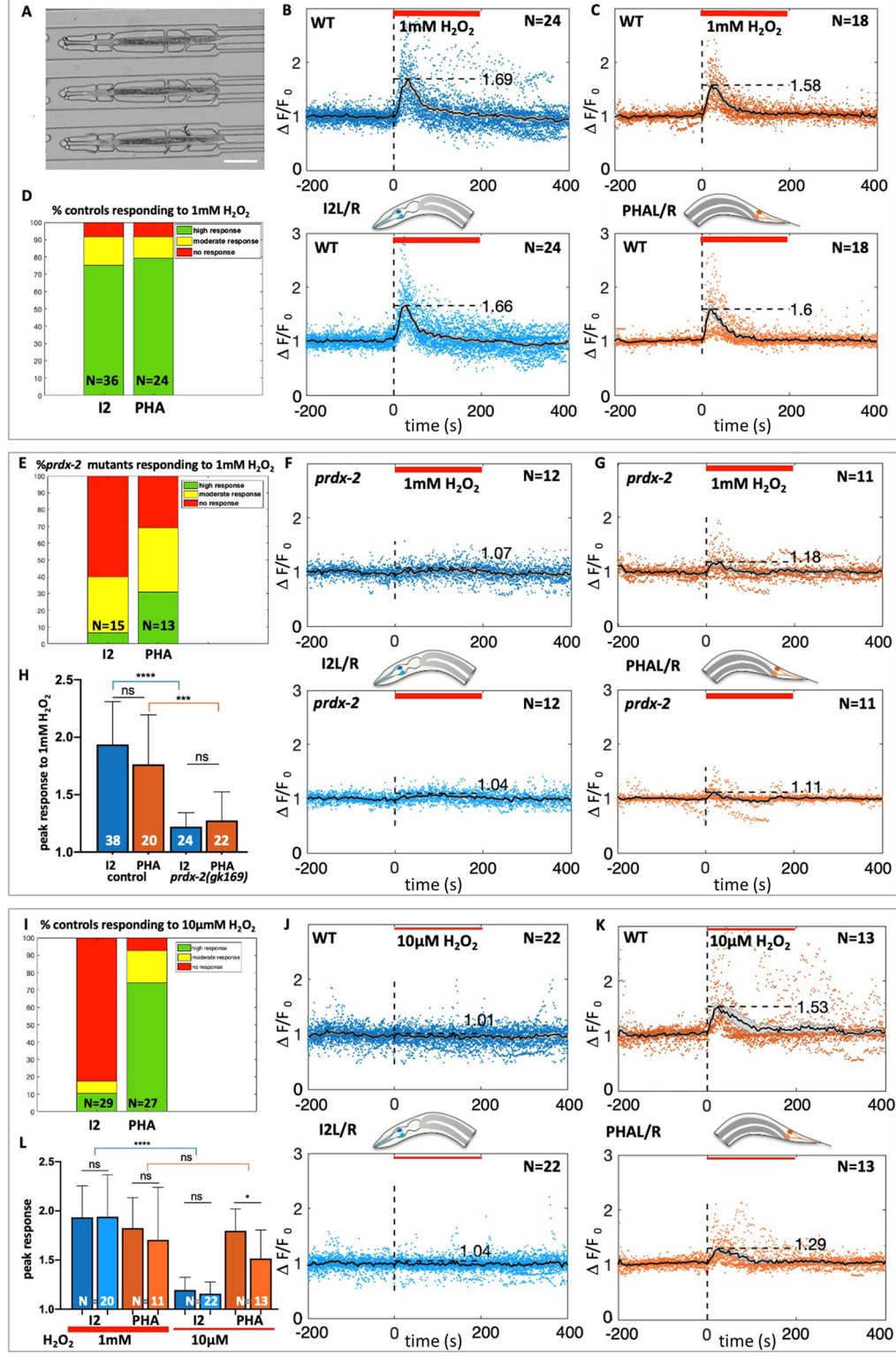

**Fig 3. I2 and PHA neurons both respond to 1mM H$_2$O$_2$ in a *prdx-2*-dependent manner, but only PHA neurons respond to 10μM H$_2$O$_2$.** (A) Low-magnification DIC picture of L4 animals trapped in the microfluidic device used in all neuron recordings experiments. (B,C,F,G,J,K) Calcium response (visualized by GCaMP3) in I2R/I2L (B,F,J) and in PHAL/PHAR neurons (C,G,K) following H$_2$O$_2$ treatment (indicated by the red bar). Average curves show normalized neuron responses (time in seconds), in both left and right neurons (top and bottom curves). N, number of movies

quantified for each genotype, in wild-type (B,C,J,K) and in *prdx-2* mutants (F,G). (D,E,I) Bar graph showing the fraction of animals responding to the H$_2$O$_2$ stimulation in all experiments, classified as high (green), moderate (yellow) or absent (red) responses (see Methods). N, number of movies analyzed. (H,L) Quantification of the calcium response to H$_2$O$_2$ in I2 and PHA neurons in controls and in *prdx-2* mutants at 1mM (H), and in I2L/R and PHAL/R controls, at 1mM and 10µM H$_2$O$_2$ (L). Bars represent the mean (N, number of movies) and error bars SD. ns, not significant, p>0.05; *p<0.05; ***p<0.001; ****p<0.0001 (Mann Whitney and Kruskal-Wallis tests). See corresponding S1–S6 Movies and S5 Fig.

PRDX-2 reproducibly shows an asymmetric expression level in I2L and I2R (Fig 1B and 1D and detailed description in [46]), each neuron was scored individually to take into account a putative left-right effect. However, this did not impact neuron response as no significant difference in the normalized response of left and right neurons was noticed (Fig 3B and 3L). Importantly, we observed that PHA neurons respond to 1mM H$_2$O$_2$ comparably to I2 neurons (Fig 3C, 3D and 3H and S2 Movie and S5 Fig). We then investigated whether PRDX-2 function was necessary for PHA response, using a strong *prdx-2* loss of function mutant. In agreement with previous work [8], I2 neurons in *prdx-2(gk169)* mutants failed to respond to 1mM H$_2$O$_2$ (Fig 3E, 3F and 3H and S3 Movie and S5 and S6 Figs). Interestingly, we uncovered that PHA neuron response was severely impaired in *prdx-2* mutants, although not completely abolished as in I2s (Fig 3E, 3G and 3H and S4 Movie and S5 and S6 Figs). From these observations, we conclude that both I2s and PHAs responses to oxidative stress require the function of the peroxiredoxin PRDX-2, but the residual response of PHA neurons suggests a lesser requirement of the antioxidant in tail neurons.

## I2s and PHAs show differences in H$_2$O$_2$ sensitivity and in receptors involved

Given the putative role of PRDX-2 as an H$_2$O$_2$ sensor, the fact that there was a slight difference in PRDX-2 activity requirement in I2s and PHAs prompted us to analyze whether the head and tail neurons share the same sensitivity to H$_2$O$_2$. We thus tested whether I2 and PHA neurons exhibit a response to the very mild dose of 10µM H$_2$O$_2$, a dose which induces a less penetrant pharyngeal pumping inhibition ($\approx$ 35% of animals) than that observed at 1mM ($\approx$ 90% of animals, [8]). In our experiments, whereas I2 neurons failed to be activated in most animals at 10µM H$_2$O$_2$ (25/29, Fig 3I, 3J and 3L and S5 Movie and S5 Fig), PHA neurons responded in the vast majority of animals (25/27, Fig 3I), in a similar manner than at 1mM H$_2$O$_2$ (Fig 3K and 3L and S6 Movie and S5 Fig). In addition, PHA neurons response to 10µM H$_2$O$_2$ depends on PRDX-2, as *prdx-2* mutants PHA neurons all fail to respond to this low dose (13/13, S7 Movie and S5 and S6 Figs). The requirement of PRDX-2 for PHA response to micromolar H$_2$O$_2$ suggests that PRDX-2 is unlikely to transmit the H$_2$O$_2$ signal under its hyperoxidized form. Taken together, we conclude that PHA neurons are more sensitive to low doses of H$_2$O$_2$ than I2 neurons.

The difference in sensitivity between I2 and PHA neurons may come from distinct molecular mechanisms. To explore this possibility, we first tested which receptors are required in I2s and in PHAs for H$_2$O$_2$ perception. We focused on photoreceptors as photosensation is likely to involve the generation of ROS [8, 47]. In *C. elegans*, light sensing relies on unusual gustatory G-protein-coupled receptors (GPCRs) related to vertebrate photoreceptors: the two nematode closest paralogs LITE-1 and GUR-3 mediate photosensation in ASJ and ASH neurons [47–49], and light and H$_2$O$_2$ sensing in I2 neurons [8]. We investigated whether these two receptors are differentially localized in I2 and PHA neurons. We generated a knock-in GUR-3::GFP line, which revealed that GUR-3 is solely expressed in I2 and I4 photosensory neurons (Fig 4A), as previously reported using episomal expression [8]. Such a restricted pattern deeply contrasts

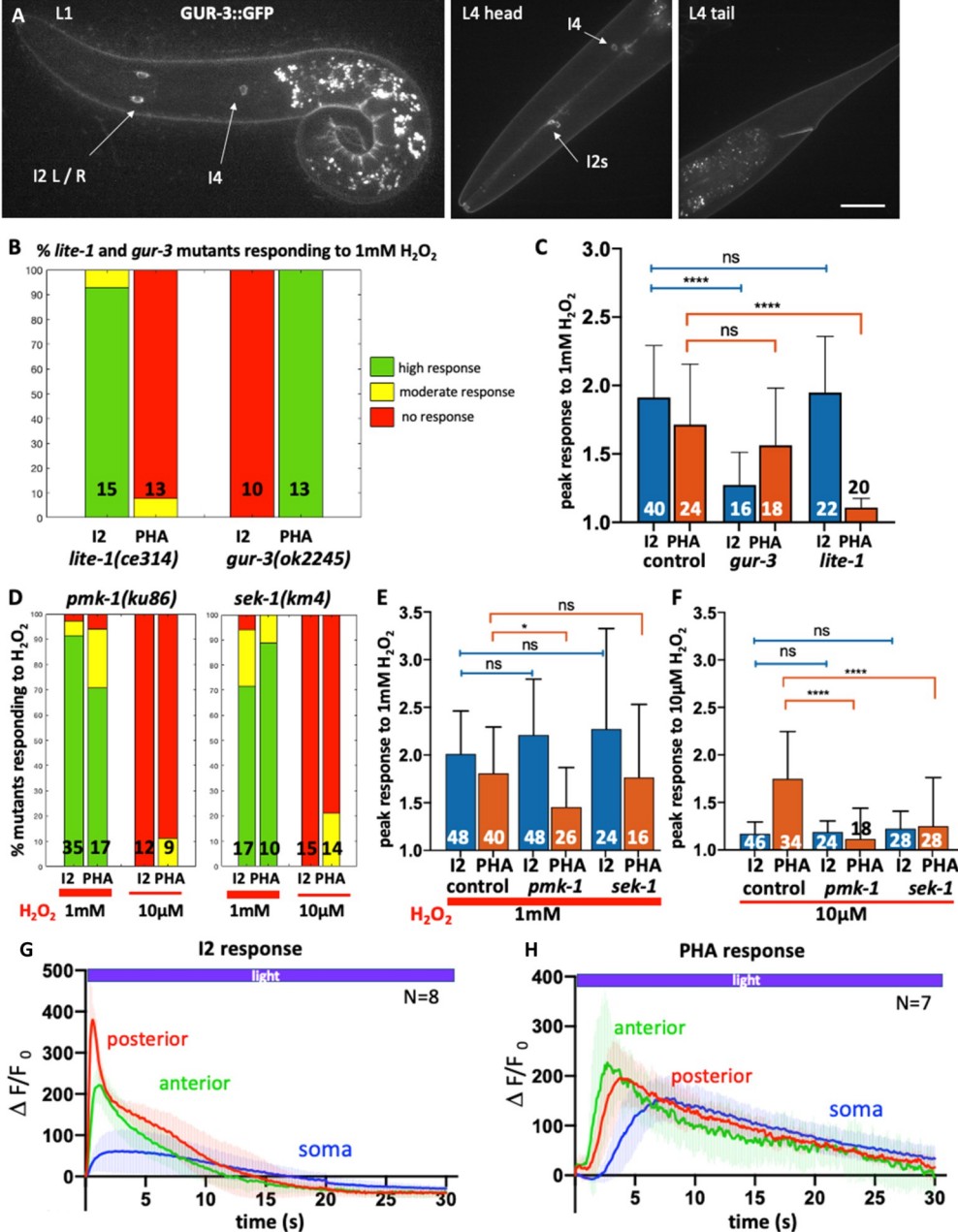

**Fig 4. H₂O₂ response involves different receptors and transducers in I2 and PHA neurons.** (A) Spinning-disc confocal projections of a representative GUR-3::GFP knock-in animal, at L1 (left) and L4 stages (right). The GUR-3:: GFP signal is detected in I2 neurons and in a single I4 neurons (head panel), but not in PHA/PHB neurons (tail panel). Bar, 10μm. (B,D) Bar graph of the fraction of *lite-1*, *gur-3*, *pmk-1* and *sek-1* mutants responding to the H₂O₂ stimulation, classified as high (green), moderate (yellow) or absent (red) responses. N, number of movies analyzed. (C, E,F) Quantification of the calcium response to 1mM H₂O₂ in I2 and PHA neurons in controls and in *gur-3* and *lite-1* mutants (C), and in *pmk-1* and *sek-1* mutants at 1mM (E) and at 10μM H₂O₂ (F). Bars represent the mean (number of movies indicated) and error bars SD; ns, not significant, p>0.05; *p<0.05; ****p<0.0001 (ANOVA and Kruskal-Wallis tests). See corresponding S8–S19 Movies and S7–S10 Figs. (G,H) Blue light triggered different calcium fluxes in I2 and PHA neurons in the three regions analyzed; anterior neurite (green), posterior neurite (red) and soma (blue). The curves represent the mean and shaded error bars indicate SD. See S20–S22 Movies.

with the broad expression domain of LITE-1, which includes phasmid neurons (PHA and PHB), but not I2 neurons, as shown using both translational and transcriptional reporters [8].

These differential localizations prompted us to inquire whether mutants in these receptors were still able to trigger a response to 1mM $H_2O_2$ in I2 and PHA neurons. In *gur-3(ok2245)* mutants, only PHA neurons were able to respond to 1mM $H_2O_2$ (Fig 4B and 4C and S8 and S9 Movies and S7 and S8 Figs), providing evidence that GUR-3 function is not essential in PHA neurons for $H_2O_2$ sensing. In contrast, *lite-1(ce314)* mutants showed a reciprocal response, with only I2 neurons responding to 1mM $H_2O_2$ (Fig 4B and 4C and S10 and S11 Movies and S7 and S8 Figs). In conclusion, $H_2O_2$ likely activates I2 neurons via *gur-3* and PHA neurons via its paralog *lite-1*, respectively. Interestingly, these observations might explain the previously reported impaired avoidance response of *lite-1* mutants to 1mM $H_2O_2$ [8], as PHA neurons are involved in escape behavior [13].

## PMK-1 and SEK-1 are required for PHA neurons response to micromolar $H_2O_2$ but are dispensable in I2 neurons

Since $H_2O_2$ perception in I2 and PHA neurons involves different receptors and requires the function of PRDX-2 in both cases, we wondered what type of signaling occurs downstream PRDX-2 to trigger neuronal activation upon $H_2O_2$ stimulation.

As mentioned above, studies in various models have reported that peroxiredoxins can modulate the p38/MAPK signaling pathway to influence cellular decisions, notably in drosophila and mammalian cells [34]. In *C. elegans*, specifically, the activation of this PRDX-2-PMK-1/ p38MAPK cascade allows micromolar doses of $H_2O_2$ to potentiate the ASH neuron sensory behavior to glycerol [3]. Therefore, we investigated whether the p38/MAPK pathway could be involved in $H_2O_2$ sensing by analyzing the responses of *pmk-1* (p38/MAPK) and *sek-1* (MAPKK) mutants in I2 and PHA neurons. As the strongest allele *pmk-1(ok811)* is homozygous lethal, we had to use the hypomorphic *pmk-1(km25)* viable mutant, which carries a N-terminal deletion [50]. In *pmk-1(km25)* mutants, while I2 neurons responded normally to 1mM $H_2O_2$, PHA neurons showed a slightly milder response to 1mM $H_2O_2$ compared to controls (Fig 4D and 4E and S12 and S13 Movies and S9 Fig). In *sek-1(km4)* MAPKK mutants, we observed comparable responses of I2 and PHA neurons to 1mM $H_2O_2$ to that of controls (Fig 4D and 4E and S16 and S17 Movies and S10 Fig). We conclude that for 1mM $H_2O_2$ sensing, the p38/MAPK pathway is dispensable at least in I2 neurons, but may play some role in PHA neurons.

In ASH neurons, the PRDX-2-mediated activation of the p38/PMK-1 cascade induces a potentiation of their sensory behavior [3]. Therefore, we asked whether the p38/PMK-1 pathway could similarly promote PHA higher sensitivity to $H_2O_2$. We examined PHA neurons response to 10μM $H_2O_2$ in *pmk-1* (MAPK) and in *sek-1* (MAPKK) mutants. Strikingly, both mutants displayed a very similar phenotype, with an abolished response of PHA neurons to micromolar doses of $H_2O_2$ observed in *pmk-1* (Fig 4D and 4F and S15 Movie and S9 Fig) and in *sek-1* mutants (Fig 4D and 4F and S19 Movie and S10 Fig). In I2 neurons, as in controls, both *pmk-1* and *sek-1* failed to respond to 10μM $H_2O_2$ (Fig 4D and 4F and S14 and S18 Movies and S9 and S10 Figs), as expected. Taken together, this suggests that the p38/MAPK pathway would be specifically required for PHA neurons hypersensitivity to $H_2O_2$, but dispensable in I2 neurons.

## PHA neurons are photosensory neurons like ASH neurons

Light sensing has been reported for ASJ, ASH and I2 neurons and require either the LITE-1 or the GUR-3 receptor, respectively [8, 47–49]. As PHAs neurons require LITE-1 to respond to $H_2O_2$ (Fig 4B and 4C and S11 Movie and S7 Fig), we asked whether they could respond to light. To test this, we monitored calcium transients in three neuronal compartments using the

GCaMP strain upon stimulating neurons with blue light, as previously done for I2 or ASH neurons [8, 47]. Interestingly, we found that all regions of PHA neurons responded to light, displaying a different response profile than those of I2 neurons (S22 Movie): PHA soma showed a stronger and longer response than I2 soma; PHA posterior neurite responded much slower that in I2 where it exhibits the fastest and strongest response peak, and the anterior neurite also had a slower recovery than in I2 (Fig 4G and 4H and S20 and S21 Movies). Overall, while I2 neurons exhibit a fast photoresponse within 10-15s, PHA neurons photoresponse requires twice longer to return to steady state (approx. 30s). Strikingly, we noticed that PHA neurons profile is highly reminiscent of that reported in ASH neurons photoresponse [47].

It has been proposed that light sensing may involve intracellular $H_2O_2$ release and peroxiredoxin signaling in both nematodes and yeast [7, 8]. To test whether PHA response to light requires the peroxiredoxin PRDX-2, we analyzed light sensing in *prdx-2* mutants. As in $H_2O_2$ sensing (Fig 3E, 3F and 3H and S3 Movie), we observed that I2 response to light strictly depends on *prdx-2* (15/15, S23 Movie and S11 Fig). Surprisingly, we found that PHA photoresponse does not require PRDX-2, as *prdx-2* mutants were still able to respond to light (17/20, S24 Movie and S11 Fig). We conclude that light sensing does not involve the same mechanisms in I2 and in PHA neurons, downstream the photoreceptors.

## Discussion

### $H_2O_2$ sensing in head and tail neurons relies on different mechanisms

Here, we describe how two pairs of sensory neurons located in the head and the tail of *C. elegans*, namely I2s and PHAs, contribute to exogenous $H_2O_2$ and light sensing. Compared to previous reports, our study relies on a PRDX-2::GFP knock-in line more closely reflecting endogenous expression level, in comparison to overexpression often observed with extrachromosomal arrays. While classical methods do not enable precise control of the environment such as application of a stress, we carried out neuron response experiments using the microfluidic technology, allowing live imaging of immobilized animals upon simultaneous exposure to a controlled oxidative stress.

We found that PHA tail neurons can elicit a response to a micromolar range of $H_2O_2$, whereas I2 head neurons cannot, suggesting that distinct molecular mechanisms may account for this difference. Accordingly, while the peroxiredoxin PRDX-2 is essential for $H_2O_2$ sensing in both I2 and PHA neurons, a different transmembrane receptor is required to transduce the signal: I2 neurons use the gustatory receptor GUR-3, while PHA neurons would require its paralogue LITE-1. Another difference lies in the p38/MAPK activity requirement: while dispensable in I2 neurons, it would be specifically required in PHA neurons to confer their hypersensitivity to $H_2O_2$ (lack of response of PHA neurons to 10μM $H_2O_2$ in *sek-1*/MAPKK and in *pmk-1*/MAPK mutants, Fig 4). Finally, although nematodes were known for a long time to avoid light when the light pulse was applied on their tail [48], our work provide the first evidence, to our knowledge, that PHA tail neurons act as photoreceptor cells.

Overall, our data are consistent with previous findings unveiling the existence of two distinct modes of response to oxidative stress in *C. elegans*: a direct response in peripheral tissues such as the gut, and a neuronally-regulated response relying on synaptic transmission [51]. Specifically, we uncovered that a harsh oxidative stress (10mM $H_2O_2$) triggers PRDX-2 induction in the anterior gut and in the EPC (Figs 1 and 2), while lower doses of $H_2O_2$ triggers either I2 and PHA neurons activation (1mM), or only PHA neurons activation (10μM) (Fig 3). Interestingly, both types of response involve the peroxiredoxin PRDX-2, which behaves differently in the two cellular contexts: PRDX-2 would be cell-autonomously induced by SKN-1 in the intestine, but likely not in neurons (Fig 2). Therefore, we propose that PRDX-2 might act as a peroxidase in

the gut, as proposed [27], whereas it could function as a $H_2O_2$-signaling molecule in neurons, as suggested for I2 neurons [8]. Importantly, $H_2O_2$ response in I2 and PHA neurons requires the joint function of PRDX-2 and a receptor, as each mutant individually cannot respond (Figs 3 and 4). In addition, former rescue experiments indicated that light response requires the activity of PRDX-2 and GUR-3 specifically in I2 neurons [8]. Based on all these data and recent studies shedding light on how $H_2O_2$ is sensed in plant and animal cells [34, 52], we propose new hypotheses regarding $H_2O_2$ signaling in neurons which are depicted in Fig 5 and described below, to integrate our findings with recent results from the literature.

## A presumptive model of $H_2O_2$ sensing in C. elegans neurons

In both I2 and PHA neurons, we favor the hypothesis that cytosolic PRDX-2 rather than the transmembrane receptor would be the neuronal $H_2O_2$ sensor, based on the following observations: i) in many cases, $H_2O_2$ signaling is mediated by oxidation of cysteines in redox-regulated proteins [53]. Alternatively, redox signaling often relies on peroxiredoxins acting as sensor and transducer of $H_2O_2$ signal [54], as thiol modifications would be much faster when catalyzed by peroxiredoxins [55–57], due to their abundance and their high reactivity to $H_2O_2$ [58]. Here, the striking abundance of PRDX-2 in PHA, and especially in I2 neurons reinforces this idea, as well as the defective response of *prdx-2* mutants to both doses of $H_2O_2$ tested (Figs 3 and S6). Consequently, we propose that $H_2O_2$ signaling in I2 and PHA neurons would involve redox signaling through PRDX-2. ii) Concerning receptor topology, GUR-3 and LITE-1 present a higher conservation in their intracellular domains, with conserved cysteines only

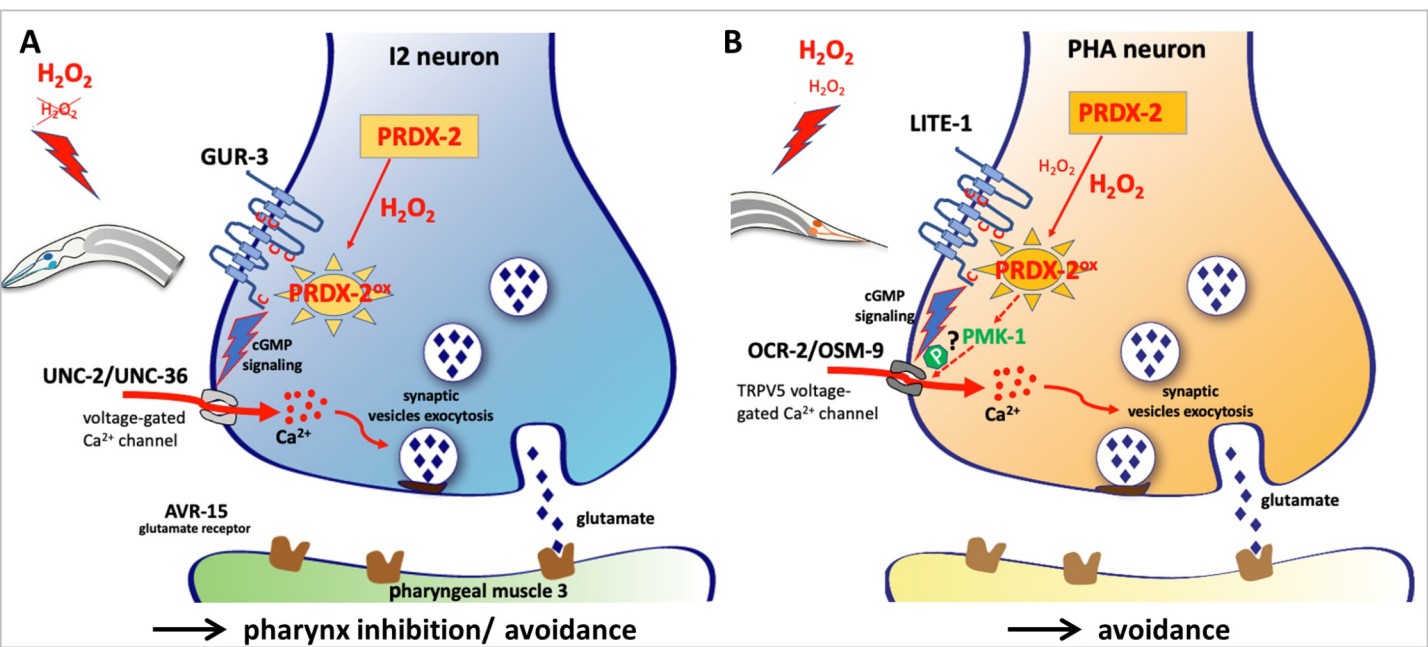

**Fig 5. Hypothetical model of $H_2O_2$ sensing and signaling in *C. elegans* I2 and PHA neurons based on this and on previous work.** (A-B) Hypothetical model of $H_2O_2$-induced neuronal activation, based on our data and on previous studies. Schematic drawing of a presynaptic button in I2 (A) and PHA (B) neurons, illustrating the presumptive $H_2O_2$-PRDX-2-mediated neuronal activation in both cases (modelled in S12 Fig). High doses of $H_2O_2$ (1mM) are sensed by both neurons, but only PHA neurons respond to 10μM $H_2O_2$, as illustrated in red. In this model, $H_2O_2$ would freely diffuse through the neuron plasma membrane and oxidize PRDX-2, presumably leading to LITE-1 or GUR-3 activation. Receptor activation is likely relayed by cGMP signaling, resulting in the opening of voltage-gated calcium channels (in grey) and neurotransmitter release (glutamate in both cases), triggering an adapted response. In PHA neurons, the PMK-1/p38MAPK pathway is additionally required to promote neuronal response to micromolar doses of $H_2O_2$, potentially through OSM-9 phosphorylation, as observed in ASH neurons [3]. See Discussion for further details and bibliographic references.

found intracellularly and in a transmembrane domain (S12A Fig). This structure is more reminiscent of that of the vertebrate transmembrane protein GDE2, which is activated intracellularly by the Prdx1 peroxiredoxin [59], than of the plant H$_2$O$_2$ sensor, the HCPA1 receptor, whose activation involves direct oxidation of extracellular cysteines [52]. Here, GUR-3 and LITE-1 receptor topology does not support the hypothesis of a direct oxidation by H$_2$O$_2$ on the extracellular domain, but rather suggests intracellular signaling as in the case of GDE2 (S12B and S12C Fig). We thus propose a scenario in which H$_2$O$_2$ would diffuse through the neuron plasma membrane and oxidize PRDX-2. Oxidized PRDX-2 or its disulfide form (PRDX-2$^{ox/S-S}$) would in turn react with the cysteines of GUR-3 or LITE-1, triggering receptor activation, and I2 or PHA neuron response (Figs 5 and S12). These hypotheses will require future experimental validation.

Neuronal response involves the opening of calcium channels, which are different in I2 and PHA neurons: I2 activation depends on UNC-2 and UNC-36 voltage-gated calcium channels [9], while PHA neurons require the cyclic nucleotide-gated channel TAX-4 and the vertebrate TRPV5 nematode equivalent OSM-9 [12]. Of note, the other TRPV5 channel subunit OCR-2 may also act in PHAs, as it is expressed in PHAs (S13 Fig, [60]) and it functions together with OSM-9 [61]. In conclusion, although closely related and both able to sense H$_2$O$_2$ and light, GUR-3 and LITE-1 receptor signaling likely involves different downstream transducers. Except for PRDX-2 requirement, I2 and PHA neurons would use distinct molecular pathways to transduce H$_2$O$_2$ response.

## A striking parallel between PHA and ASH neurons

Our data uncover a high sensitivity of PHA neurons to micromolar doses of H$_2$O$_2$ that is not seen in I2 neurons and requires PRDX-2 and p38/MAPK activity. Strikingly, micromolar doses of H$_2$O$_2$ were also reported to activate the p38/MAPK pathway via PRDX-2 in ASH neurons, leading to the phosphorylation of the OSM-9 TRPV sensory channel, thereby increasing its sensitivity [3]. Although our study and the latter do not elucidate how PRDX-2 triggers PMK-1 activation, recent evidence sheds light on this process: in both mammalian and drosophila cells, H$_2$O$_2$ induces transient disulfide-linked conjugates between the MAP3K and a typical 2-Cys peroxiredoxin [34]. Similarly, in *C. elegans*, PRDX-2 could activate the MAPKKK NSY-1, as NSY-1 function is required in ASH neuron [3], and may be expressed in PHAs (*unidentified cells of the tail*, *D. Moerman*, *WormBase*). Based on these observations and on the fact that OSM-9 is expressed [62] and likely required in PHA neurons [12], it is possible that PMK-1/p38MAPK activation might increase PHA neurons' sensitivity to H$_2$O$_2$ through the downstream phosphorylation of OSM-9, via AKT-1, as in ASH neurons (Fig 5). Alternatively, by analogy with recent work showing that DLK/p38MAPK signaling controls LITE-1 stability in ASH neurons, PMK-1 could also control LITE-1 turnover by RAB-5-mediated endocytosis [63]. These assumptions, which will necessitate further experimental testing, are supported by the observation that ASHL/R and PHAL/R are both descendants of ABplp or ABprp in the nematode cell lineage, and hence may express similar sets of genes following lineage-specific priming [64].

The fact that PHA, PHB and ASH were found in the same neuron cluster [65] confirms that they share similar molecular signatures. To illustrate the parallel between ASH and PHA polymodal nociceptors, we retrieved the set of genes expressed in these neurons [65], and analysed which ones were specifically enriched in these neurons in comparison to all other neurons (S13 Fig, data in S1–S3 Tables). This analysis revealed that ASH and PHA not only express many genes specific to ciliated neurons, as expected (*eg. che-3*, *nphp-4*, *che-11*, *ifta-1*, C33A12.4 and R102.2, [66]), but also a number of common receptors (*eg. ocr-2*, *osm-9*, *ida-1*,

*casy-1*, *ptp-3* or *pdfr-1*), although some of them are more enriched in ASH (*dop-2*, *nrx-1*, *snt-5*, *sue-1*). Intriguingly, their neuropeptide profile appears different for the two neuron pairs (PHA enriched in *flp-7*, *flp-4*, *flp-16*, *nlp-1 ins-18*, while ASH strongly express *flp-13* and *npr* genes), suggesting that beyond their functional similarity, ASH and PHA may trigger different types of intercellular communication. Finally, this analysis highlights the fact that several genes with unknown function are highly enriched in both ASH and PHA (*eg.* F27C1.11, W05F2.7, *tos-1*, *cab-1*), pointing to their potential role in neuronal function.

Taken together, our analyses highlight the common features shared between PHA and ASH polymodal nociceptive neurons, as formerly noticed [13], since they both: i) display a higher sensitivity dependent on the p38/MAPK pathway (Fig 4D–4F and [3]), ii) exhibit a similar response profile to light (Fig 4H and [47]), iii) require the photoreceptor LITE-1 for lightsensing [47] or H$_2$O$_2$ sensing (Fig 4B and 4C), iv) trigger avoidance [13], and v) share a close molecular signature (S13 Fig and [65]). A recent report indicates that ASJ neurons are also required for H$_2$O$_2$ avoidance [6], and unravels the behavioral mechanisms allowing *C. elegans* to find a suitable niche, owing to the interplay between H$_2$O$_2$ and bacteria in its environment. Overall, these observations led us to propose that nematodes might integrate the environmental redox signals from at least four different pairs of neurons (I2, ASH, PHA and ASJ) in order to trigger an appropriate dose-dependent physiological response.

## GUR-3 and LITE-1 receptors mediate both light and H$_2$O$_2$ sensing

In yeast, light sensing relies on the peroxisomal oxidase Pox1 which triggers light-dependent H$_2$O$_2$ formation, the latter being sensed by the Tsa1 peroxiredoxin and transduced to thioredoxin for subsequent signaling [7]. Nematodes, unlike yeast, require a photoreceptor in addition to the antioxidant for lightsensation: GUR-3 in I2 neurons [8], and LITE-1 in ASJ and ASH neurons [47, 49]. Here we showed that both I2 and PHA respond to light, but with a different profile. Despite its unusual membrane topology, LITE-1 has been shown to encode a bona fide photoreceptor, whose photoabsorption depends on its conformation [67]. However, whether light directly activate the neuron photoreceptor or triggers intracellular H$_2$O$_2$ release and signaling is still unclear. In agreement with the fact that H$_2$O$_2$ inhibits LITE-1 photoabsorption *in vitro* [67], it has been shown that a H$_2$O$_2$ pretreatment reduces LITE-1-mediated photoresponse in ASH neurons [47]. Consistent with these reports, our data illustrate that GUR-3 and LITE-1 have a dual function in both light and H$_2$O$_2$ sensing. In I2 neurons, as in yeast, redox signaling could be involved in transducing the light signal, as PRDX-2 is strictly required for light sensing (S11 Fig, [8]). In contrast, PHA neurons can respond to light without PRDX-2, indicating that the LITE-1 photoreceptor tranduces the light signal without a redox relay. This difference could explain our observation that I2 neurons respond faster to light than PHA and ASH neurons (Fig 4G and 4H and [47]). Whether this difference strictly depends on the photoreceptor (LITE-1 in PHA, ASH *vs* GUR-3 in I2) and/or on its downstream signaling cascade is an open question.

Finally, it is noteworthy that I2 and PHA neurons rely on peroxiredoxin for H$_2$O$_2$ and/or light sensing, while ASJ and ASH rely on thioredoxin: TRX-1 is required for LITE-1-dependent photosensation in ASH neurons [47], and is expressed in ASJ photosensory neurons [68], which respond to H$_2$O$_2$ [6]. Altogether, this further underlines the importance of a redox signaling relay involving antioxidants of the peroxiredoxic cycle in nematode photosensory neurons.

In conclusion, our work illustrates that nematodes can sense various concentrations of H$_2$O$_2$ through sensory neurons located in the head and the tail, using either partially different (I2 and PHA), or similar molecular mechanisms (ASH and PHA). This set of neurons also

confer light sensing to the nematode with a distinct speed in the response (fast in I2, slow in PHA and ASH, [47]). While I2 neurons seem more specialized in sensing oxidative stress [8, 10], PHA and ASH can detect many other stimuli [12], but all of them can trigger avoidance. To tackle the complex question of how these neuronal inputs translates into behavior, a recent method was developed [69], allowing the simultaneous recording of behavioral and neural responses of *C. elegans* to salt concentrations changes. These observations may help uncover whether and how inputs from head and tail oxidative stress sensory neurons integrate to allow nematodes to quickly and appropriately react to a change in the environment.

## Experimental procedures

### Generation of plasmids and transgenic strains by CRISPR/Cas9-mediated genome editing

*C. elegans* strains (listed in Supplementary Information) were maintained as described [70]. PRDX-2::GFP knock-in strain was generated by CRISPR/Cas9-mediated genome editing, using a DNA plasmid-based repair template strategy [37]. For both PRDX-2 and GUR-3 knock-ins, a C-terminal GFP fusion was generated, comprising a flexible linker between the coding region and GFP to allow correct folding of the fusion protein. A combined small guide-RNA/repair template plasmid was built using the SAP Trap strategy [71]. Phusion DNA polymerase was used to amplify by PCR 5' and 3' *prdx-2* and *gur-3* homology arms (HAs) from N2 genomic DNA, using primers containing SapI restriction sites and silent mutations to prevent Cas9 re-cleavage (all primers sequences listed in Supplementary Information). After purification, 5' and 3' HAs and sgRNA oligonucleotides were assembled into the destination vector (pMLS256), together with the flexible linker (from pMLS287), GFP and the *unc-119* rescuing element (from pMLS252), in a single SapI restriction-ligation reaction, as described [71]. Prior to transformation in DH5α cells, a sabotage restriction was performed with SpeI to digest empty destination vectors but not the desired assembly constructs, which were subsequentially verified by restriction digest analysis and sequencing. All plasmids used for injection were purified using a DNA Miniprep Kit (PureLink, Invitrogen), or a DNA midi-prep kit (Macherey Nagel). For PRDX-2::GFP knock-in, a plasmid mix containing combined sgRNA/repair template plasmid (50 ng/μl), Cas9-encoding pSJ858 (25ng/μl) and co-injection markers (pCFJ90 at 2.5ng/μl; pCFJ104 at 5ng/μl, and pGH8 at 5ng/μl) was injected in the germline of *unc-119(ed3)* animals [37]. For GUR-3::GFP knock-in, an injection mix containing purified Cas9 protein (IDT) associated with tracrRNA and crRNA (guide RNA), GUR-3 repair template, and co-injection markers was injected in *unc-119(ed3)* animals, according to IDT online protocols for *C. elegans*. Plates containing 2/3 injected F0 animals were starved, chunked on fresh plates, for candidates screening (attested by the presence of wild-type non fluorescent animals). Knock-in events were validated by PCR on homozygous lysed worms (QuantaBio AccuStart II GelTrack PCR SuperMix), using primers annealing in the inserted sequence and in an adjacent genomic region not included in the repair template. The PRDX-2::GFP strain was outcrossed 5 times to N2 wild-types.

### RNA interference

RNAi experiments were performed by feeding using the Ahringer-MRC feeding library [72]. Animals fed with the empty vector L4440 served as a negative control. The efficiency of each RNAi experiment was assessed by adding an internal positive control, *zyg-9(RNAi)*, which induces embryonic lethality.

## Spinning-disk confocal microscopy acquisitions and fluorescence intensity measurements

For live imaging, animals were anesthetized in M9 containing 1mM levamisole and mounted between slide and coverslip on 3% agarose pads. Synchronized L4 animals were treated for 30min in a 96-well flat bottom plate, in 50µl of M9 containing 1mM or 10mM H$_2$O$_2$. Treated animals were transferred using a siliconized tip on a freshly seeded plate to recover, and imaged 1h30 to 2h later. Spinning-disk confocal imaging was performed on a system composed of an inverted DMI8 Leica microscope, a Yokogawa CSUW1 head, an Orca Flash 4.0 camera (2048*2018 pixels) piloted by the Metamorph software. Objective used were oil-immersion 40X (HC PL APO, NA 1.3) or 63X (HCX PL APO Lambda blue, NA 1.4). The temperature of the microscopy room was maintained at 20˚C for all experiments. Z-stacks of various body regions were acquired with a constant exposure time and a constant laser power in all experiments. Maximum intensity projections were used to generate the images shown. Fluorescence intensity measurements in int1, I2 and EPC cells were performed using the Fiji software, by manually drawing a region of interest (ROI) around the cell (int1, EPC), or applying a threshold (I2 neurons), background was subtracted and average pixel intensity was quantified.

## Microfabrication and microfluidic chip preparation

The microfluidic chip original design was inspired by the Wormspa [73], but pillars distances were adapted to trap L4 animals, and multiple series of traps were included to increase the number of experiments per chip (S4 Fig). A master mold was made by standard soft photolithography processes by spin-coating a 25µm layer of SU-8 2025 (Microchem, USA) photoresist at 2700 rpm for 30sec on a 3" wafer (Neyco, FRANCE). Then, we used a soft bake f 7min at 95˚C on hot plates (VWR) followed by a UV 365nm exposure at 160 mJ/cm$^2$ with a mask aligner (UV-KUB3 Kloé®, FRANCE). Finally, a post-exposure baking identical to the soft bake was performed before development with SU-8 developer (Microchem, USA). Then, the wafer was baked at 150˚C for 15min to anneal potential cracks and strengthen the adhesion of the resist to the wafer. Finally, the master mold was treated with chlorotrimethylsilane to passivate the surface.

Worm microchannels were cast by curing PDMS (Sylgard 184,10:1 mixing ratio), covalently bound to a 24 × 50 mm coverslip after plasma surface activation (Diener, Germany), and incubated 20min at 60˚C for optimal adhesion. The chip was perfused with filtrated M9 solution through the medium inlet using a peristaltic pump (Ismatec), until complete removal of air bubbles. Worm loading was performed with the pump set at a low flow rate (<30 µl/min), through a distinct inlet (S4 Fig): 10–15 young L4 animals (synchronized by bleaching 48h prior to each experiment) were picked in a siliconized Eppendorf tube (Sigmacote SL2, Sigma Aldrich) containing M9, and perfused into the traps. The loaded chip was carried to the microscope while being still connected to the pump by a gravity flow (preventing animals to escape) until the microfluidic chip was installed on the microscope stage.

## Calcium imaging

I2 and PHA neuronal response was monitored using the calcium sensor GCaMP3 expressed under the *flp-15* promoter as in [8]. To image H$_2$O$_2$ response, young L4 animals trapped in microfluidic chips were imaged using the confocal spinning disc system described above with the 20X air objective (HC PL APO CS2, NA 0.75). The microfluidic chip allowed the simultaneous recording of up to 3 animals per experiment (S4 Fig). Z-stacks of 10–15 images (10 µm

spacing) were acquired every 2s (using the stream Z mode), for 350 time points. Exposure time was 50ms and laser power set on 40%. The device was perfused with M9 medium throughout the experiment using a peristaltic pump set at 80 μl/min, and $H_2O_2$ was perfused (at 10μM or 1mM in M9) for 100 time points (3min20s) after an initial recording of 35–45 time points. Movies were computationally projected using MetaMorph, and data processing (including movie registration, neuron segmentation and tracking over time) was conducted with a custom-developed Matlab program detailed below and available at https://github.com/gcharvin/viewworm (a user guide is provided in Supplementary Information).

To image light response in I2 and PHA neurons, L4 worms were mounted on a slide covered with 3% agarose pads in M9 supplemented with 1mM levamisole. Video-recordings were performed on the spinning-disc microscope using the 40X oil objective. Animals were exposed to blue light (485nm) while their neuronal response was simultaneously recorded in stream mode (10 frames/sec, single Z, 100ms exposure, laser 100%) for 30sec.

## Calcium response analyses

For $H_2O_2$ response analyses, sequences of images were spatially realigned with respect to the first image of the timeseries in order to limit the apparent motion of the worm in the trap and ease the tracking of neurons of interest. This image registration process was performed using standard 2D image cross-correlation by taking the first image as a reference. Then, we used a machine-learning algorithm (based on a decision tree) to segment pixels in the fluorescent images. For this, we took a series of image transforms (gaussian, median, range filters) as descriptors for the classifier, and we trained the model on typically 10 frames before applying the result to the rest of the time series. This segmentation method appeared to be superior to simple image thresholding, which is inadequate when dealing with fluorescent signals that vary both in time and space (the brightness of two neurons is quite different). Next to the segmentation procedure, we tracked the identified neurons using distance minimization, and we quantified the mean fluorescence signal in each neuron over time. Last, fluorescence data corresponding to individual animals were pooled after synchronization from the time of exposure to $H_2O_2$ and signal normalization. This image analysis pipeline is available at https://github.com/gcharvin/viewworm and a tutorial for use is included in Supplementary Information. As some movies could not be quantified due to uncontrolled animal movements, a visual classification of neuronal responses (high, moderate, absent) was made by comparison to successfully tracked movies.

For light response analyses in I2 and PHA neurons, the same image processing pipeline as in [8] was used, except that ROI were manually drawn in Fiji.

## Statistical analyses

For pairwise comparisons of data sets with a normal distribution (or N>30), p-values were calculated using an unpaired two-tailed Student test, and the Welch correction was applied when samples variance was not homogeneous. When distributions were not normal (Anderson-Darling and Shapiro Wilk tests not satisfied), a Mann Whitney test was used. Multiple comparisons were analyzed using a one-way ANOVA with Bonferroni's correction (for normal distributions), or a non-parametrical Kruskal-Wallis test followed by a Dunn's multiple comparisons test. Statistical analyses were conducted with the GraphPad Prism9 software.

Mean are represented and error bars indicate standard deviation (SD) in all figures. The data presented here come from at least three independent experiments. For p values, not significant p>0.05; *p<0.05; **p<0.01; ***p<0.001: ****p<0.0001.

## Supporting information

**S1 Fig. Related to Fig 1- The available PRDX-2::GFP transgenic line (CTD1051.3) does not suit to our study.** (A) Low magnification fluorescent image of untreated animals from the CTD1051.3 line showing that transgenics express aggregates of PRDX-2::GFP, a hallmark of overexpression of the fusion protein. (B,C) Low magnification images of animals treated in flat bottom wells imaged after 24h of treatment in the potent oxidative stress inducing agent tBOOH. In the wild-type control well (A), almost all animals are all dead in 2mM tBOOH, appearing as rods, while CTD1051.3 transgenics survive a 25X higher dose of the drug, indicating their much stronger resistance to oxidative stress. (D) Representative image of an animal of the PRDX-2::GFP knock-in line; a wild-type control imaged with the same settings is shown (E), delineated by a dotted line. Bar, 100μm. (F-H) Confocal projections showing the tail region (dotted contours) of a PRDX-2::GFP transgenic animal stained with the lipophilic orange-red dye DiI (see https://www.wormatlas.org/EMmethods/DiIDiO.htm), imaged in green (F) and red channels (G). The overlay (H) shows that PRDX-2::GFP tail neurons are stained by the dye, establishing their identity as phasmid sensory neurons (PHA/PHB). Bar, 10μm.
(TIF)

**S2 Fig. Related to Figs 1 and 2- PRDX-2 induction is not observed in I2 neurons upon H$_2$O$_2$ treatment.** (A,B) Wild-type controls do not show a higher gut autofluorescence after H$_2$O$_2$ treatment, as illustrated by the int1 anterior gut cell fluorescence quantification (C). (D, E) Quantification of the PRDX-2::GFP fluorescence level in I2 neurons and in the excretory pore cell (EPC) in controls after 1mM-H$_2$O$_2$ treatment. (F,G) Quantification of I2 neurons' PRDX-2::GFP fluorescence level in *skn-1(zj15)* mutants and in *skn-1(RNAi)* animals upon a 10mM-H$_2$O$_2$ treatment. Means are shown and error bars represent SD; ns, not significant, p>0.05 (t test or Mann-Whitney test). Scale bar, 50μm.
(TIF)

**S3 Fig. Related to Fig 2.** Genomic organization of the *prdx-2* locus and presumptive regulation by SKN-1 and additional transcription factors. (A) Genome Browser screenshot from WormBase (release WS283, JBrowse II), showing the gene organization on chromosome II, at the indicated coordinates (top), and the peaks detected by ChIP-seq using an anti-GFP antibody [42] in *prdx-2* and *pkc-3* promoters, in GFP-tagged transgenic lines of the indicated transcription factors (B). Note the co-regulation of *prdx-2* and *pkc-3*, which are organized in an operon, illustrated by the green bar.
(TIF)

**S4 Fig. Related to Figs 3 and 4- Design of the microfluidic chip used in H$_2$O$_2$ neuron response experiments.** (A) Global view of the microfluidic chip used in all neuron response experiments, showing the 6 independent series of 10 worm traps. The original file (Autocad format) is available at https://github.com/gcharvin/viewworm. (B) Magnification of the worm trap area showing the 10 individual channels (corresponding to the red box in A). Scale bar, 1mm. (C) DIC image of an animal trapped (anterior to the left). Occasionally animals were trapped in the opposite orientation, but this did not affect the neuronal response. Scale bar, 100μm.
(TIF)

**S5 Fig. Related to Fig 3- Individual intensity measurements of S1–S7 Movies.** The curves represent the mean GCaMP3 intensity raw value over time (1 frame = 2sec) quantified in I2 and PHA neurons in control and *prdx-2* mutants corresponding movies (S1–S7 Movies), upon

a 1mM or a 10μM $H_2O_2$ exposure. T0 indicate the time point at which the $H_2O_2$ treatment has been applied during 100 frames. Red and blue colors represent left and right I2 and PHA neurons (*ie* I2L/R and PHAL/R). Note that the colors have been changed in normalized average curves shown in Fig 3 and in related S6 Fig.
(TIF)

**S6 Fig. Related to Fig 3- *prdx-2* mutants PHA neurons do not respond to 10μM $H_2O_2$.**
(A-D) Average curves showing the normalized calcium response to 10μM $H_2O_2$ measured over time (in seconds) using the GCaMP3 sensor in PHA left and right neurons (top and bottom curves) in *prdx-2(gk169)* mutants (B) and in wild-type controls (A). N, number of movies analyzed for each genotype. See S6–S7 Movies and related S5 Fig.
(TIF)

**S7 Fig. Related to Fig 4- *gur-3* and *lite-1* mutants show reciprocal phenotypes in $H_2O_2$ sensing in I2 and in PHA neurons.** (A-D) Average curves showing the normalized calcium response to 1mM $H_2O_2$ measured over time (indicated in seconds) using the GCaMP3 sensor in I2 and PHA left and right neurons (top and bottom curves) in *gur-3(ok2245)* (A,B) and *lite-1(ce314)* mutants (C,D). N, number of movies analyzed for each genotype. See S8–S11 Movies and related S8 Fig.
(TIF)

**S8 Fig. Related to Fig 4- Individual intensity measurements of S8–S11 Movies (*gur-3* and *lite-1* mutants).** The curves represent the mean GCaMP3 intensity raw value over time (1 frame = 2sec) quantified in I2 and PHA neurons in *gur-3* and *lite-1* mutants upon a 1mM $H_2O_2$ exposure (starting at T0 and lasting 100 frames), in corresponding movies (S8–S11 Movies). Red and blue colors show left and right neurons for I2 (left panel) and PHA (right panel) neurons (*ie* I2L/R and PHAL/R). Note the reciprocal phenotypes observed in both mutants. Colors have been changed in the related normalized average curves shown in S7 Fig.
(TIF)

**S9 Fig. Related to Fig 4, continued- Individual intensity measurements of S12–S15 Movies (*pmk-1* mutants).** The curves represent the mean GCaMP3 intensity raw value over time (1 frame = 2sec) quantified in I2 and PHA neurons in *pmk-1* mutants upon a 1mM or a 10μM $H_2O_2$ stimulation (starting at T0 and lasting 100 frames), in corresponding movies (S12–S15 Movies). Red and blue color indicate I2L/R neurons (left panel) and PHAL/R neurons (right panel).
(TIF)

**S10 Fig. Related to Fig 4, continued- Individual intensity measurements of S16–S19 Movies (*sek-1* mutants).** The curves represent the mean GCaMP3 intensity raw value over time (1 frame = 2sec) quantified in I2 and PHA neurons in *sek-1* mutants upon a 1mM or a 10μM $H_2O_2$ stimulation (starting at T0 and lasting 100 frames), in corresponding movies (S16–S19 Movies). Red and blue color indicate left and right I2 (left panel) and PHA (right panel) neurons.
(TIF)

**S11 Fig. Related to Fig 4, continued—Unlike in I2 neurons, PRDX-2 is not essential for light sensing in PHA neurons.** Average curves showing the normalized calcium response to blue light over time (in seconds) using the GCaMP3 sensor, measured in the soma of I2 (A) and PHA neurons (B) in wild-type controls and in *prdx-2(gk169)* mutants. N, number of movies analyzed for each genotype. Error bars represent SD. While I2 neurons fail to respond to light in *prdx-2* mutants, PHA neurons do respond, albeit with a lower intensity peak than in controls. See related S20–S24 Movies.
(TIF)

**S12 Fig. Related to Discussion- A presumptive model of how GUR-3 and LITE-1 receptors may be activated intracellularly by PRDX-2.** (A) Alignment of LITE-1 and GUR-3 protein sequences made with the SIM alignment tool (https://web.expasy.org/sim/), using the comparison matrix BLOSUM30. Transmembrane (TM) and intra/extracellular domains were predicted using the DeepTMHMM program (https://dtu.biolib.com/DeepTMHMM). The alignment reveals 39.9% identity over 434 residues overlap. Note the rather large intracellular domains (in blue), encompassing conserved cysteines (boxed in red) (B-C) A putative PRDX-2 redox relay may trigger $H_2O_2$-induced receptor activation in I2 and PHA neurons. Sketch of GUR-3 (B) and LITE-1 (C) receptors, deduced from A, depicting their conserved cysteines. Upon $H_2O_2$ exposure, oxidized PRDX-2 or its disulfide form (PRDX-2$^{ox/S-S}$) could oxidize these cysteines, possibly forming a disulfide conjugate and/or inducing a conformation change, which would subsequently trigger receptor activation, and I2 or PHA neuron response.
(TIF)

**S13 Fig. Related to Discussion- PHA and ASH neurons belong to the same neuronal cluster, and share partially overlapping transcriptomic signatures.** (A) Uniform Manifold Approximation and Projection (UMAP) projection of parent cluster 28, from [65] in which ASH and PHA/PHB nociceptive neurons were found in distinct sub-clusters (Louvain clustering at 8 PCs and at 1.2 resolution). (B, C) Dot plot indicating for a selection of genes both the intensity of gene expression and the fraction of expressing cells in each sub-cluster (B), based on single-cell RNA-sequencing data from [38]. See Supplementary Information and related S1–S3 Tables for exhaustive lists of genes expressed in each cluster.
(TIF)

**S14 Fig.**
(TIF)

**S1 Movie. (Related to Fig 3)- Wild-type I2 response to 1mM $H_2O_2$.** On-chip 4D movie showing control I2 neurons response to 1mM $H_2O_2$ visualized using the GCaMP3 calcium sensor. The movie shown corresponds to a Z-projection of all time points (acquired every 2 sec), which has been processed for re-alignment (using the Matlab Readworm_PHA code). Accelerated 60 times. See fluorescence intensity measurements of this movie in S5 Fig.
(MP4)

**S2 Movie. (Related to Fig 3)- Wild-type PHA response to 1mM $H_2O_2$.** On-chip 4D movie showing control PHA neurons response to 1mM $H_2O_2$ visualized using the GCaMP3 calcium sensor. The movie shown corresponds to a Z-projection of all time points (acquired every 2 sec), which has been processed for re-alignment (using the Matlab Readworm_PHA code). Accelerated 60 times. See fluorescence intensity measurements of this movie in S5 Fig.
(MP4)

**S3 Movie. (Related to Fig 3)- *prdx-2* mutant I2 response to 1mM $H_2O_2$.** On-chip 4D movie showing the absence of I2 neurons response to 1mM $H_2O_2$ in a *prdx-2* mutant. The movie shown corresponds to a Z-projection of all time points (acquired every 2 sec), which has been processed for re-alignment (using the Matlab Readworm_PHA code). Accelerated 60 times. See fluorescence intensity measurements of this movie in S5 Fig.
(MP4)

**S4 Movie. (Related to Fig 3)- *prdx-2* mutant PHA response to 1mM $H_2O_2$.** On-chip 4D movie showing the absence of PHA neurons response to 1mM $H_2O_2$ in a *prdx-2* mutant. The movie shown corresponds to a Z-projection of all time points (acquired every 2 sec), which

has been processed for re-alignment (using the Matlab Readworm_PHA code). Accelerated 60 times. See fluorescence intensity measurements of this movie in S5 Fig.
(MP4)

**S5 Movie. (Related to Fig 3)- Wild-type I2 response to 10μM H₂O₂.** On-chip 4D movie showing the absence of I2 neurons response to 10μM H₂O₂ in a control animal. The movie shown corresponds to a Z-projection of all time points (acquired every 2 sec), which has been processed for re-alignment (using the Matlab Readworm_PHA code). Accelerated 60 times. See fluorescence intensity measurements of this movie in S5 Fig.
(MP4)

**S6 Movie. (Related to Fig 3)- Wild-type PHA response to 10μM H₂O₂.** On-chip 4D movie showing control PHA neurons response to 10μM H₂O₂ visualized using the GCaMP3 calcium sensor. The movie shown corresponds to a Z-projection of all time points (acquired every 2 sec), which has been processed for re-alignment (using the Matlab Readworm_PHA code). Accelerated 60 times. See fluorescence intensity measurements of this movie in S5 Fig.
(MP4)

**S7 Movie. (Related to S6 Fig)- *prdx-2* mutant PHA response to 10μM H₂O₂.** On-chip 4D movie showing the absence of PHA neurons response to 10μM H₂O₂ in a *prdx-2* mutant. The movie shown corresponds to a Z-projection of all time points (acquired every 2 sec), which has been processed for re-alignment (using the Matlab Readworm_PHA code). Accelerated 60 times. See fluorescence intensity measurements of this movie in S5 Fig.
(MP4)

**S8 Movie. (Related to Fig 4)- *gur-3* mutant I2 response to 1mM H₂O₂.** On-chip 4D movie showing the absence of I2 neurons response to 1mM H₂O₂ in a *gur-3* mutant. The movie shown corresponds to a Z-projection of all time points (acquired every 2 sec), which has been processed for re-alignment (using the Matlab Readworm_PHA code). Accelerated 60 times. See fluorescence intensity measurements of this movie in S8 Fig and related average response in S7 Fig.
(MP4)

**S9 Movie. (Related to Fig 4)- *gur-3* mutant PHA response to 1mM H₂O₂.** On-chip 4D movie showing PHA neurons response to 1mM H₂O₂ in a *gur-3* mutant. The movie shown corresponds to a Z-projection of all time points (acquired every 2 sec), which has been processed for re-alignment (using the Matlab Readworm_PHA code). Accelerated 60 times. See fluorescence intensity measurements of this movie in S8 Fig and related average response in S7 Fig.
(MP4)

**S10 Movie. (Related to Fig 4)- *lite-1* mutant I2 response to 1mM H₂O₂.** On-chip 4D movie showing I2 neurons response to 1mM H₂O₂ in a *lite-1* mutant. The movie corresponds to a Z-projection of all time points (acquired every 2 sec), which has been processed for re-alignment (using the Matlab Readworm_PHA code). Accelerated 60 times. See fluorescence intensity measurements of this movie in S8 Fig and related average response in S7 Fig.
(MP4)

**S11 Movie. (Related to Fig 4)- *lite-1* mutant PHA response to 1mM H₂O₂.** On-chip 4D movie showing the absence of PHA neurons response to 1mM H₂O₂ in a *lite-1* mutant. The movie shown corresponds to a Z-projection of all time points (acquired every 2 sec), which has been processed for re-alignment (using the Matlab Readworm_PHA code). Accelerated 60

times. See fluorescence intensity measurements of this movie in S8 Fig and related average response in S7 Fig.
(MP4)

**S12 Movie. (Related to Fig 4)-** *pmk-1* **mutant I2 response to 1mM H₂O₂.** On-chip 4D movie showing I2 neurons response to 1mM H₂O₂ in a *pmk-1* mutant. The movie shown corresponds to a Z-projection of all time points (acquired every 2 sec), which has been processed for re-alignment (using the Matlab Readworm_PHA code). Accelerated 60 times. See fluorescence intensity measurements of this movie in S9 Fig.
(MP4)

**S13 Movie. (Related to Fig 4)-** *pmk-1* **mutant PHA response to 1mM H₂O₂.** On-chip 4D movie showing PHA neurons response to 1mM H₂O₂ in a *pmk-1* mutant. The movie shown corresponds to a Z-projection of all time points (acquired every 2 sec), which has been processed for re-alignment (using the Matlab Readworm_PHA code). Accelerated 60 times. See fluorescence intensity measurements of this movie in S9 Fig.
(MP4)

**S14 Movie. (Related to Fig 4)-** *pmk-1* **mutant I2 response to 10µM H₂O₂.** On-chip 4D movie showing the absence of I2 neurons response to 10µM H₂O₂ in a *pmk-1* mutant. The movie shown corresponds to a Z-projection of all time points (acquired every 2 sec), which has been processed for re-alignment (using the Matlab Readworm_PHA code). Accelerated 60 times. See fluorescence intensity measurements of this movie in S9 Fig.
(MP4)

**S15 Movie. (Related to Fig 4)-** *pmk-1* **mutant PHA response to 10µM H₂O₂.** On-chip 4D movie showing the absence of PHA neurons response to 10µM H₂O₂ in a *pmk-1* mutant. The movie shown corresponds to a Z-projection of all time points (acquired every 2 sec), which has been processed for re-alignment (using the Matlab Readworm_PHA code). Accelerated 60 times. See fluorescence intensity measurements of this movie in S9 Fig.
(MP4)

**S16 Movie. (Related to Fig 4)-** *sek-1* **mutant I2 response to 1mM H₂O₂.** On-chip 4D movie showing I2 neurons response to 1mM H₂O₂ in a *sek-1* mutant. The movie shown corresponds to a Z-projection of all time points (acquired every 2 sec), which has been processed for re-alignment (using the Matlab Readworm_PHA code). Accelerated 60 times. See fluorescence intensity measurements of this movie in S10 Fig.
(MP4)

**S17 Movie. (Related to Fig 4)-** *sek-1* **mutant PHA response to 1mM H₂O₂.** On-chip 4D movie showing PHA neurons response to 1mM H₂O₂ in a *sek-1* mutant. The movie shown corresponds to a Z-projection of all time points (acquired every 2 sec), which has been processed for re-alignment (using the Matlab Readworm_PHA code). Accelerated 60 times. See fluorescence intensity measurements of this movie in S10 Fig.
(MP4)

**S18 Movie. (Related to Fig 4)-** *sek-1* **mutant I2 response to 10µM H₂O₂.** On-chip 4D movie showing the absence of I2 neurons response to 10µM H₂O₂ in a *sek-1* mutant. The movie shown corresponds to a Z-projection of all time points (acquired every 2 sec), which has been processed for re-alignment (using the Matlab Readworm_PHA code). Accelerated 60 times. See fluorescence intensity measurements of this movie in S10 Fig.
(MP4)

**S19 Movie. (Related to Fig 4)-** *sek-1* **mutant PHA response to 10μM H₂O₂.** On-chip 4D movie showing the absence of PHA neurons response to 10μM H₂O₂ in a *sek-1* mutant. The movie shown corresponds to a Z-projection of all time points (acquired every 2 sec), which has been processed for re-alignment (using the Matlab Readworm_PHA code). Accelerated 60 times. See fluorescence intensity measurements of this movie in S10 Fig.
(MP4)

**S20 Movie. (Related to Fig 4)- Wild-type control I2 response to light.** Single-Z movie (recorded at 10 fps) showing control I2 neurons response to blue light (485nm), visualized with the GCaMP3 calcium sensor. The original movie has been colored artificially to highlight fluorescence intensity variations, using Fiji 'glow' look up table. Note the quasi instantaneous and very strong posterior neurite response. Accelerated 5 times.
(AVI)

**S21 Movie. (Related to Fig 4)- Wild-type control PHA response to light.** Single-Z movie (recorded at 10 fps) showing control PHA neurons response to blue light (485nm), visualized with the GCaMP3 calcium sensor. The original movies has been colored artificially to highlight fluorescence intensity variations, using Fiji 'glow' look up table. Note the long-lasting PHA soma response. Accelerated 5 times.
(AVI)

**S22 Movie. (Related to Fig 4)- Wild-type control I2 and PHA simultaneous responses to light.** Single-Z movie (recorded at 10 fps) showing simultaneously control I2 (right) and PHA (left) neurons response to blue light (485nm). The original movies has been colored artificially to highlight fluorescence intensity variations, using Fiji 'glow' look up table. Note the contrast of response, with the quasi instantaneous response of I2 neurons versus the slower and longer soma response in PHA neurons. Accelerated 5 times.
(AVI)

**S23 Movie. (Related to Fig 4)-** *prdx-2* **mutant I2 response to light.** Single-Z movie (recorded at 10 fps) showing the absence of I2 neurons response to blue light (485nm) in a *prdx-2* mutant, visualized with the GCaMP3 calcium sensor. The original movie has been colored artificially to highlight fluorescence intensity variations, using Fiji 'glow' look up table. Accelerated 5 times. See related average response in S11 Fig.
(AVI)

**S24 Movie. (Related to Fig 4)-** *prdx-2* **mutant PHA response to light.** Single-Z movie (recorded at 10 fps) showing PHA neurons response to blue light (485nm) in a *prdx-2* mutant, visualized with the GCaMP3 calcium sensor. The original movie has been colored artificially to highlight fluorescence intensity variations, using Fiji 'glow' look up table. Note the long-lasting PHA soma response in *prdx-2* mutant as in WT (S21 Movie). Accelerated 5 times. See related average response in S11 Fig.
(AVI)

**S1 Graphical abstract.**
(TIF)

**S1 File.**
(PDF)

**S2 File.**
(PDF)

**S1 Table.**
(XLSX)

**S2 Table.**
(XLSX)

**S3 Table.**
(XLSX)

## Acknowledgments

We are grateful to all the staff members of the Imaging Center of the IGBMC, especially Elvire Guiot, Marine Silvin, Erwan Grandgirard and Bertrand Vernay for assistance in confocal microscopy. We thank Christelle Gally, Basile Jacquel and Eric Marois for helpful discussions and critical reading of the manuscript, Sandra Bour for assistance with figure design, Doulaye Dembele for help with statistic analyses. We are indebted to the Reymann, Vermot and Jarriault labs for sharing their equipments and reagents, and for scientific input. We thank the Horvitz lab, especially Na An, for providing the MT GCaMP strains and the Pujol lab for providing the *sek-1* mutant. We thank WormBase for providing resources and the *Caenorhabditis* Genetics Center (funded by NIH Office of Research Infrastructure Programs P40 OD010440, University of Minnesota) for providing strains.

## Author Contributions

**Conceptualization:** Sophie Quintin.

**Data curation:** Sophie Quintin, Tao Ye.

**Formal analysis:** Sophie Quintin.

**Funding acquisition:** Gilles Charvin.

**Methodology:** Théo Aspert.

**Resources:** Tao Ye, Gilles Charvin.

**Software:** Gilles Charvin.

**Writing – original draft:** Sophie Quintin.

**Writing – review & editing:** Théo Aspert, Gilles Charvin.

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
