## [Decision Letter · Decision Letter 0]

29 Apr 2022

PONE-D-22-08767Distinct mechanisms underlie H2O2 sensing in C. elegans head and tailPLOS ONE

Dear Dr. QUINTIN,

Thank you for submitting your manuscript to PLOS ONE. After careful consideration, we feel that it has merit but does not fully meet PLOS ONE’s publication criteria as it currently stands. Therefore, we invite you to submit a revised version of the manuscript that addresses the points raised during the review process. I encourage you to consider the comments of each review, with particular focus on some of the questions of reviewer 1. Specifically, certain aspects require more explanation, including how the identity of neurons was determined. This reviewer also raised some concerns about the statistical analyses and questions whether t-test are always the right test. Please also consider the suggestion by reviewer 2 to genetically dissect, using mutants or RNAi, the mechanism by which pmk-1 is required for H2O2 sensing.

We look forward to receiving your revised manuscript.

Kind regards,

Elise A. Kikis

Academic Editor

PLOS ONE

Journal Requirements:

[We are grateful to all the staff members of the Imaging Center of the IGBMC, especially Elvire Guiot, Erwan Grandgirard and Bertrand Vernay for assistance in confocal microscopy. We thank Christelle Gally, Basile Jacquel and Eric Marois for helpful discussions and critical 

reading of the manuscript, and Sandra Bour for assistance with figure design. We are indebted to the Reymann, Vermot and Jarriault labs for sharing their equipments and reagents. We thank the Horvitz lab, especially Na An, for providing the MT GCaMP strains. We thank 

WormBase and the Caenorhabditis Genetics Center (funded by NIH Office of Research Infrastructure Programs P40 OD010440, University of Minnesota) for providing strains.]

 [This work was funded by the grant ANR-10-LABX-0030-INRT, a French state fund of the Agence Nationale de la Recherche, attributed to Gilles Charvin.]

Reviewers' comments:

Reviewer's Responses to Questions

**Comments to the Author**

1. Is the manuscript technically sound, and do the data support the conclusions?

Reviewer #1: Partly

Reviewer #2: Yes

2. Has the statistical analysis been performed appropriately and rigorously? 

Reviewer #1: No

Reviewer #2: Yes

3. Have the authors made all data underlying the findings in their manuscript fully available?

Reviewer #1: Yes

Reviewer #2: Yes

4. Is the manuscript presented in an intelligible fashion and written in standard English?

Reviewer #1: No

Reviewer #2: Yes

5. Review Comments to the Author

Reviewer #1: The work of Quintin et al. examines the role of hydrogen peroxide sensing at the head and tail of C. elegans. The I2 pharyngeal neurons in the head and phasmid PHA neurons in the tail can both respond to H2O2, but with different sensitivities. The PHA neurons are more sensitive to micromolar concentrations whereas the I2 neurons require milimolar levels. The peroxiredoxin PRDX-2 is required in both cases, but may regulate detoxification in the I2 neurons and ROS signaling in the PHA neurons. The receptors GUR-3 and LITE-1 were found to play a role in light sensing and hydrogen peroxide sensing in the I2 neurons and PHA neurons, respectively. The paper provides insight into how animals can respond to the same signal through different mechanisms. Overall the experiments add to the field but there are some areas for improvement.

The authors measure changes in prdx-2::GFP expression in response to H2O2 in different cell types. How did the authors confirm the localization/identity of the neurons?

The concentrations of H2O2 differ experiments and range from 10mM, 1mM and 10uM. Please provide rationale for changing the concentrations in each experiment. Also, a discussion on the biological relevance of these concentrations would be helpful for the reader.

Statistical analysis should be improved. For some figures, a t-test is not appropriate. Welsh’s t-test should be corrected to “Welch”

Figure 1 – The authors show an increase in signal following exposure to H2O2 in the intestinal cells, which the authors attribute to increased expression of prdx-2::GFP. However, it appears that the time hours of 1-2 hours may not be sufficient to induce protein translation and the signaling could be an artifact of gut granule autofluorescence in response to the H2O2. The authors should demonstrate the selectivity of the single for prdx-2. In addition, the supplemental data in WT control should be quantified (2A and B).

From Figure 1, the authors make conclusions on the role of prdx-2 in I2 neurons based on expression changes (or lack thereof). While I agree that the responses of prdx-2 to oxidative stress are most likely “cellular context dependent”, the author’s conclusion is solely based on a GFP signal, which does not report on prdx-2 activity. Thus the authors should not overstate their conclusion.

The article has grammatical issues and unfinished sentences. As a note, the supplemental tables are not represented as tables. Genomic nomenclature should be followed throughout – sometimes the gene is listed when referring to the protein and vice versa. Also some of the titles do not fit with the data. One example, “Expression pattern of prdx-2 and its evolution upon H2O2 treatment.” How is prdx-2 evolving with H2O2 exposure? I would recommend editing throughout.

Figure 2 – The wormbase analysis seems more suited for the supplement since this is not new data. Moreover, it is unclear if the authors are permitted to reuse the data in their figure.

Figure 5 – The hypothetical model provided in C and D are helpful and supported by the data provided. However, A and B do not add any value and are confusing since the authors did not test or discuss any specific cysteine residues in their protein target. Additionally, it appears that the cysteine residues do not match with the protein sequences provided on wormbase.

The discussion is very detailed and often meanders from discussing the findings of the manuscript. I would suggest condensing the discussion to avoid over interpretation.

Reviewer #2: The manuscript submitted by Quintin et al identified how oxidants such as hydrogen peroxide (H2O2)are perceived by the I2 pharyngeal neurons and the tail phasmid PHA neurons in C. elegans. Using a GFP knock-in line for the peroxiredoxin PRDX-2, the authors demonstrate this peroxiredoxin functions as a putative H2O2 senor in both I2 and PHA neurons, while in the anterior gut and excretory pore cell it functions mainly as a peroxidase. Furthermore, the activation of the I2 and PHA neurons were dependent on the concentration of H2O2 administered which led to the identification of distinct molecular mechanisms to explain differences in sensitivity. Gustatory G-protein-coupled receptors GUR-3 and LITE-1 were shown to be expressed and required for the perception of H2O2 in the I2 and PHA neurons respectively. Further investigation using microfluidics and calcium imaging revealed PHA neurons required the p38 MAPK pathway in sensing H2O2 which has been shown to activate the voltage gated calcium channel OSM-9 in ASH neurons in the worm. Finally, in addition to sensing H2O2 the PHA neurons exhibited a slow photo-response to light, while the I2 neurons exhibited a fast response. The photo-response is dependent on PRDX-2 in the I2 neurons. However, in the PHA neurons PRDX-2 is not required. The manuscript is well written and addresses an important question in the field.

Concerns

1. It is thought that dimerized peroxiredoxin can interact with ASK-1 (NSY-1) to activate the p38 MAPK. Using calcium imaging the authors show pmk-1 is required for the sensing of H2O2 and suggest the p38 MAPK pathway is involved in this mechanism. However, the sek-1 and the nsy-1 mutant strains weren't included in the study. Including calcium measurements in the sek-1 and nsy-1 mutant strains would further support this claim.

2. According to the Li et al (2016) AKT-1 acts downstream of the p38 MAPK to phosphorylate OSM-9 in ASH neurons in response to H2O2 stimulation. Does Akt-1 act downstream of PMK-1 to activate OSM-9 in the PHA neurons? This could be tested using calcium imaging or speculated in the model. Furthermore, SKN-1 is also known to act downstream of the p38 MAPK. Could SKN-1 be a potential downstream activator if AKT-1 is not required in H2O2 sensing?

6. PLOS authors have the option to publish the peer review history of their article (what does this mean?). If published, this will include your full peer review and any attached files.

Reviewer #1: No

Reviewer #2: No

---

## [Author Response · Author response to Decision Letter 0]

10 Aug 2022

Rebuttal Letter on Plos One Manuscript #PONE-D-22-08767

Summary of revisions

We thank the reviewers for their very careful evaluation of our manuscript. We took into consideration their comments, which we found all relevant. Specifically, the reviewers suggested to perform complementary analyses and controls, including more appropriate statistical analyses than a systematic use of t-test. To address Reviewer 2’s major concern, we added new data: we analyzed another mutant of the p38MAPK pathway, the sek-1 MAPKKK mutant. Performing these analyses further strengthened our conclusion that the p38MAPK pathway is required in PHA neurons for sensitivity to low doses of H2O2 (see new Fig 4, panels D-F). Furthermore, we systematically re-did all statistical analyses taking into account the multiple comparisons problem. Importantly, performing these more stringent statistical analyses did not change our conclusions. 

For these reasons, we think that our revised manuscript has significantly improved compared to its initial version, and we believe that it fits PLOS ONE’s publication criteria. 

Figures have been revised as follows:

— Figure 1 : panels E-F have been updated and include more data points, to reinforce the point that an increased PRDX-2::GFP signal is only observed in the gut upon a 10mM H2O2 treatment.

— Figure 2: panels C-D have been updated (multiple comparison statistical tests) and panel E has been transferred to a supplementary figure (S3 Fig), as it contained data retrieved from WormBase (addressing Reviewer 1’s point 7). 

— Figure 3: panels H and L have been updated with the multiple comparison statistical tests, and their legends have been modified accordingly (addressing R1 point 3); panel K has been updated with 2 additional movies. 

— Figure 4: panels D-F include the new data obtained with sek-1 MAPKK mutants analyses (addressing R2 point 1) and revised statistical analyses (R1 point 3).

— Figure 5: the hypothetical models from panels A-B have been moved to a supplementary figure (S12 Fig) to address R1 point 8.

Supplementary Figures have been revised as follows:

— S1 Fig includes new panels (F-H) to ascertain the identity of PRDX-2-expressing PHA/PHB neurons (addressing R1 point 1)

— S2 Fig includes a new panel (C) corresponding to the quantification of the gut autofluorescence in control animals treated with H2O2 (addressing R1 point 4), demonstrating the specificity of the PRDX-2::GFP signal.

— S3 Fig has been added, corresponding to the former bottom panel of Figure 2, showing the presumptive transcription factors binding sites in the promoter of prdx-2 (addressing R1 point 7).

— S10 Fig has been added, showing the individual fluorescence intensity measurements of the new supplementary sek-1 movies, and illustrates the lack of response of sek-1 mutants to 10μM H2O2,as observed in pmk-1 mutants (addressing R2 point 1).

— S13 Fig has been added, corresponding to the former top panel of Figure 5 (addressing R1 point 8). It also includes a new panel (A) with an alignment between GUR-3 and LITE-1 proteins, showing their conserved intracellular cysteines, further reinforcing the presumptive receptor activation model proposed in the discussion.

Supplemental material has been added, to support the new data included in the revised version (especially Figure 4):

— movie 16 (related to S10 Fig) - sek-1 mutant I2 response to 1mM H2O2

— movie 17 (related to S10 Fig) - sek-1 mutant PHA response to 1mM H2O2 

— movie 18 (related to S10 Fig) - sek-1 mutant I2 response to 10μM H2O2

— movie 19 (related to S10 Fig) - sek-1 mutant PHA response to 10μM H2O2 

S1-S3 Tables have been reorganized to address R1 point 6.

Concerning the References list, note that we have included 2 new papers: Zhang et al., 2022 (ref 62), which came out while we were working on revisions. It has been cited in the Discussion section (p. 11) for the reasons explained below (see R2 point 2 response). We have also added Quintin and Charvin (ref 73) in the Result section (p. 6), which has been submitted in the meantime and is currently under revision at micropublication Biology. This study, which describes a left-right asymmetry in I2 and PHA neurons, justified our approach to analyze individually left and right I2 and PHA neurons.

Please find below the original reviews as well as our reply/ comments in bold. 

Reviewer #1: The work of Quintin et al. examines the role of hydrogen peroxide sensing at the head and tail of C. elegans. The I2 pharyngeal neurons in the head and phasmid PHA neurons in the tail can both respond to H2O2, but with different sensitivities. The PHA neurons are more sensitive to micromolar concentrations whereas the I2 neurons require milimolar levels. The peroxiredoxin PRDX-2 is required in both cases, but may regulate detoxification in the I2 neurons and ROS signaling in the PHA neurons. The receptors GUR-3 and LITE-1 were found to play a role in light sensing and hydrogen peroxide sensing in the I2 neurons and PHA neurons, respectively. The paper provides insight into how animals can respond to the same signal through different mechanisms. Overall the experiments add to the field but there are some areas for improvement.

 We thank R1 for his globally positive evaluation of our work, and for his/her suggestions for improvement. However, we were a bit puzzled to read that R1 wrote that PRDX-2 ‘may regulate detoxification in the I2 neurons and ROS signaling in the PHA neurons’. We understand from this comment that our message was not fully clear, as we clearly think that PRDX-2 is involved in ROS signaling also in I2 neurons. We have tried to make this more clear in this revised version (see Results, page 5 and Discussion, page 9). 

Point 1- The authors measure changes in prdx-2::GFP expression in response to H2O2 in different cell types. How did the authors confirm the localization/identity of the neurons?

 We agree with R1 that neuron identification can sometimes be difficult in C. elegans. However, here, it was not the case for the reasons detailed below. 

Regarding I2 neurons: as stated in the manuscript (page 4), previous work already reported the expression pattern of a PRDX-2 reporter transgene, especially in I2 neurons (Isermann et al., 2004 ; Bhatla and Horvitz, 2015). Importantly, a functional validation of this finding has been performed by mutant rescue analyses (Bhatla and Horvitz, 2015). The authors demonstrated that the light-induced feeding inhibition defect of gur-3 and prdx-2 mutants are both rescued by specifically restoring gur-3 and prdx-2 functions in I2 neurons, using an I2-specific promoter (termed “I2::gur-3” and “I2::prdx-2”), derived from the flp-15 gene (Kim and Li, 2004). Therefore, as GUR-3 and PRDX-2 functions are required in I2 neurons, there is no doubt on the identity of these head neurons. 

Regarding CAN and PHA/PHB neurons; we indeed report for the first time expression of PRDX-2 in CAN and PHA/PHB tail neurons. The first assumptions were made by comparing the localization of PRDX-2::GFP positive neurons with neurons maps available at Wormatlas.org, as explained below.

— CAN neurons were easy to identify, as they are the only neurons positioned on each side midway along the body, labeled by PRDX-2::GFP (see Fig 1). Two lines of evidence support this observation (shown below): prdx-2 transcripts were abundantly detected in CAN neurons (Cao et al. 2017), and CAN neurons exhibit a very strong oxidized/reduced HyPer signal following exposure to H2O2 (Back et al, 2012), suggesting that these neurons are very sensitive to H2O2, as expected for PRDX-2 expressing cells.

This is indicated in the manuscript (p. 4): “Consistent with this, a high number of prdx-2 transcripts was detected in CANs and in the EPC (Cao et al, 2017).

— Regarding PHA/PHB neurons: their identification was made easier due to the fact that there are much fewer neurons in the tail than in the head. Based our initial observations, PRDX-2::GFP candidate neurons could have been DVA, DVB or DVC from the dorso-rectal ganglion, or anterior-most neurons from the lumbar ganglion, ie PHA, PHB (named phasmids) or PVQ, as illustrated below (Wormatlas images). Due to the rather posterior and ventral position of PRDX-2::GFP neurons right above the rectum slit, dorso-rectal ganglion neurons were excluded. 

To discriminate between the remaining candidates (phasmids vs PVQ interneurons), a DiI filling assay was done in the PRDX-2::GFP knock-in line (a common method in C. elegans to stain amphid and phasmid sensory neurons, which are exposed to the environment). As PRDX-2::GFP neurons were stained by the red lipophilic dye, this validated the neuronal PHA/PHB identity (see below, we added this data to S1 Fig).

As a last piece of evidence, the GcAMP reporter we used, driven by the flp-15 promoter (specific to I2 and PHA neurons, Kim and Li, 2004) marks the same cells as PRDX-2::GFP (shown below). 

In conclusion, we are certain about the identity of PRDX-2::GFP-expressing neurons, however we preferred not to include such a level of detail in the supplementary data as they already comprise 13 figures and 24 movies. As referenced on page 4, we included a new panel in S1 Fig (F-H), showing the DiI staining. 

Point 2- The concentrations of H2O2 differ experiments and range from 10mM, 1mM and 10uM. Please provide rationale for changing the concentrations in each experiment. Also, a discussion on the biological relevance of these concentrations would be helpful for the reader.

We agree with R1 that this point deserved a clarification. 

The choice of H2O2 concentrations was based on previous work, describing the animal’s response to various doses of H2O2. A sentence has been added in the manuscript (p.4) : ‘We selected H2O2 concentrations inducing different physiological responses (Bhatla and Horvitz, 2015). Interestingly, our observations regarding PRDX-2::GFP higher expression in the gut upon a 10mM treatment can find an explanation in the different behavior reported, as discussed later in the same paragraph (see p. 5, top).

Regarding neuron response analyses, we chose 1mM and 10μM again on the basis of the difference reported in animals’ physiological response: whereas at 1mM H2O2 nearly all animals stop pharyngeal pumping, the 10μM dose induces a less penetrant response, which we precisely aimed to test. We modified the sentence on p. 6 as follows:

While it was initially stated: ‘We thus tested whether I2 and PHA neurons exhibit a response to the very mild dose of 10μM H2O2. Indeed, at this dose, it has been reported that only a minority of animals (� 35%) respond by inhibiting pharyngeal pumping (Bhatla and Horvitz, 2015)’ ; the sentence was changed as follows: ‘We thus tested whether I2 and PHA neurons exhibit a response to the very mild dose of 10μM H2O2, a dose which induces a less penetrant pharyngeal pumping inhibition (� 35% of animals) than that observed at 1mM (� 90% of animals, Bhatla and Horvitz, 2015).

Point 3- Statistical analysis should be improved. For some figures, a t-test is not appropriate. 

We thank again R1 for the major improvement suggested here.

For pairwise comparisons of data sets with a normal distribution (or N>30), p-values were calculated using an unpaired two-tailed Student test, and the Welch correction was applied when samples variance was not homogeneous. When distributions were not normal (Anderson-Darling and Shapiro Wilk tests not satisfied), a Mann Whitney test was used. Multiple comparisons were analyzed using a one-way ANOVA with Bonferroni’s correction (for normal distributions), or a non-parametrical Kruskal-Wallis test followed by a Dunn’s multiple comparisons test. Importantly, performing these new statistical analyses on our data did not change the validity of our conclusions. These modifications have been indicated in the paragraph describing statistical analyses (page 15) and in Figure legends, accordingly. 

Welsh’s t-test should be corrected to “Welch”

Done. 

Point 4- Figure 1 – The authors show an increase in signal following exposure to H2O2 in the intestinal cells, which the authors attribute to increased expression of prdx-2::GFP. However, it appears that the time hours of 1-2 hours may not be sufficient to induce protein translation and the signaling could be an artifact of gut granule autofluorescence in response to the H2O2. The authors should demonstrate the selectivity of the single for prdx-2. In addition, the supplemental data in WT control should be quantified (2A and B).

Regarding the possibility that PRDX-2::GFP could respond later than the examined time point, in our initial experiments we did not observe any significant difference in PRDX-2::GFP level when animals were analyzed 3-4 hours vs 1-2 hours after H2O2 treatment. Notably, in yeast, the peroxiredoxin reporter Tsa1-GFP (equivalent of PRDX-2) shows rapid induction with a peak within the 100 first minutes following the H2O2 stress (Goulev et al., 2017). To demonstrate the specificity of the PRDX-2 GFP signal, we are providing a quantification of H2O2-treated controls, which do not exhibit higher gut granule fluorescence after 10mM H2O2 treatment, in contrast to PRDX-2::GFP animals. This control quantification panel has been added in the revised version (S2 Fig, panel C). We are grateful to R1 for suggesting this improvement, which strengthens our conclusions.

Point 5- From Figure 1, the authors make conclusions on the role of prdx-2 in I2 neurons based on expression changes (or lack thereof). While I agree that the responses of prdx-2 to oxidative stress are most likely “cellular context dependent”, the author’s conclusion is solely based on a GFP signal, which does not report on prdx-2 activity. Thus the authors should not overstate their conclusion.

We agree with R1 that this conclusion should not be overstated, relative to the actual experiments. We indicated this limitation in our paragraph conclusion in the following manner (p. 5):

‘Although the induction we observed is only based on expression level and not on protein activity, we suggest that PRDX-2 could still scavenge H2O2 in the EPC and in the foregut, consistent with the reported protective role of intestinal PRDX-2 (Oláhová et al., 2008).’

Point 6- The article has grammatical issues and unfinished sentences. As a note, the supplemental tables are not represented as tables. Genomic nomenclature should be followed throughout – sometimes the gene is listed when referring to the protein and vice versa. Also some of the titles do not fit with the data. One example, “Expression pattern of prdx-2 and its evolution upon H2O2 treatment.” How is prdx-2 evolving with H2O2 exposure? I would recommend editing throughout.

We thank again R1 for helping us to improve the quality of the writing. We have changed the paragraph title into : ‘Expression pattern of PRDX-2::GFP and its variation upon H2O2 treatment’

Genomic nomenclature has been addressed as it should be in the C. elegans field, ie with gene names in small letters and proteins in capital letters (this does not apply to yeast Yap1). We modified the supplemental material so that only true tables are named tables (ie Tables S1-S3), while strain and primers lists are no longer called Tables but just ‘lists’. In addition, the revised manuscript has been extensively proof-read.

Point 7- Figure 2 – The wormbase analysis seems more suited for the supplement since this is not new data. Moreover, it is unclear if the authors are permitted to reuse the data in their figure.

We agree with R1 that this figure panel was more appropriate in a supplementary figure, as it was not based on our own findings. As commonly practiced in the literature related to C. elegans, we incorporated a screenshot arising from our data mining of Wormbase, properly citing the people who generated the raw data (in this case, Niu et al, 2011) and acknowledging WormBase (p.16). The panel has been moved to S3 Fig.

Point 8- Figure 5 – The hypothetical model provided in C and D are helpful and supported by the data provided. However, A and B do not add any value and are confusing since the authors did not test or discuss any specific cysteine residues in their protein target. Additionally, it appears that the cysteine residues do not match with the protein sequences provided on wormbase.

We appreciated R1’s positive evaluation of our model (formerly Fig 5CD). Regarding the presumptive PRDX-2-mediated receptor activation (formerly Fig 5AB), we agree with R1 that the model is speculative, and has not been tested experimentally. Still, we think that it has conceptual value, as it was inspired by an important literature survey and discussions with experts in the yeast Redox community (especially Dr Jacquel, acknowledged p. 16). 

The PRDX-2 activation model has therefore been moved to a new supplementary figure (S12 Fig). In addition, we included a protein alignment between GUR-3 and LITE-1 receptors (panel A), to illustrate that conserved cysteines are only present in intracellular or transmembrane domains, thus supporting a presumptive intracellular activation by PRDX-2. Cysteine numbering has been corrected —we are grateful to R1 for noticing such a detail.

Point 9- The discussion is very detailed and often meanders from discussing the findings of the manuscript. I would suggest condensing the discussion to avoid over interpretation.

We condensed the discussion as much as we could, and did our best to avoid over interpretation. We especially attempted to place our interpretations in the context of a meaningful review of the relevant literature, also benefiting from the input of our Redox specialist colleagues. Thus, we feel that many of our discussion points are relevant and fill some gaps in the field. Therefore, we hope to have reached a satisfactory level of concision, while still highlighting biological connections that are important to understand this biological system.

Reviewer #2: The manuscript submitted by Quintin et al identified how oxidants such as hydrogen peroxide (H2O2)are perceived by the I2 pharyngeal neurons and the tail phasmid PHA neurons in C. elegans. Using a GFP knock-in line for the peroxiredoxin PRDX-2, the authors demonstrate this peroxiredoxin functions as a putative H2O2 senor in both I2 and PHA neurons, while in the anterior gut and excretory pore cell it functions mainly as a peroxidase. Furthermore, the activation of the I2 and PHA neurons were dependent on the concentration of H2O2 administered which led to the identification of distinct molecular mechanisms to explain differences in sensitivity. Gustatory G-protein-coupled receptors GUR-3 and LITE-1 were shown to be expressed and required for the perception of H2O2 in the I2 and PHA neurons respectively. Further investigation using microfluidics and calcium imaging revealed PHA neurons required the p38 MAPK pathway in sensing H2O2 which has been shown to activate the voltage gated calcium channel OSM-9 in ASH neurons in the worm. Finally, in addition to sensing H2O2 the PHA neurons exhibited a slow photo-response to light, while the I2 neurons exhibited a fast response. The photo-response is dependent on PRDX-2 in the I2 neurons. However, in the PHA neurons PRDX-2 is not required. The manuscript is well written and addresses an important question in the field.

We sincerely thank R2 for his/her very positive review.

Concerns

1. It is thought that dimerized peroxiredoxin can interact with ASK-1 (NSY-1) to activate the p38 MAPK. Using calcium imaging the authors show pmk-1 is required for the sensing of H2O2 and suggest the p38 MAPK pathway is involved in this mechanism. However, the sek-1 and the nsy-1 mutant strains weren't included in the study. Including calcium measurements in the sek-1 and nsy-1 mutant strains would further support this claim.

We agree with R2 that our initial conclusion on the p38MAPK pathway requirement in PHA hypersensitivity to H2O2 relied solely on the analysis of a single mutant, the pmk-1/MAPK mutant. As suggested, we further assessed the involvement of this pathway in PHA neurons by introducing the GcAMP calcium sensor in a sek-1/MAPKK mutant background (SXB70 new strain) to monitor I2 and PHA responses to H2O2. We observed that I2 neurons in sek-1 mutants responded normally to 1mM (movie 16), but not to 10μM H2O2 (movie 18), as in controls and as in pmk-1 mutants. Similarly, sek-1 mutants PHA neurons responded normally to 1mM H2O2 (movie 17), while pmk-1 mutant showed a slight diminution compared to WT. Importantly, in sek-1 mutants, as in pmk-1 mutants, PHA neurons were no longer able to respond to the low dose of 10μM H2O2 (movie 19), unlike in controls. Thus, our new analyses reveal that sek-1 mutants almost completely phenocopied pmk-1 mutants in their response to H2O2, further supporting the point that the p38MAPK pathway seems indeed to be specifically involved in PHA hypersensitivity to H2O2. 

These new results have been incorporated in the revised version of Figure 4 (panels D-F); and they are supported by the addition of supplementary material (movies 16-19 and S10 Fig, depicting their quantification). The text has been accordingly modified in the manuscript in the results sections (paragraph now entitled ‘PMK-1 and SEK-1 are required for PHA neurons response to micromolar H2O2 but are dispensable in I2 neurons’, p. 7-8) and in the discussion (p. 9). Given that the C. elegans p38-MAPK pathway is linear (Inoue et al., 2005), and that sek-1 mutants showed the same defects as PMK-1 in PHA neurons, we considered that testing additional components of the same pathway (such as the nsy-1 MAPKKK) was not necessary here. 

2. According to the Li et al (2016) AKT-1 acts downstream of the p38 MAPK to phosphorylate OSM-9 in ASH neurons in response to H2O2 stimulation. Does Akt-1 act downstream of PMK-1 to activate OSM-9 in the PHA neurons? This could be tested using calcium imaging or speculated in the model. Furthermore, SKN-1 is also known to act downstream of the p38 MAPK. Could SKN-1 be a potential downstream activator if AKT-1 is not required in H2O2 sensing?

We thank R2 for raising this hypothesis, highlighting his/her interest in our work. Li et al. (2016) indeed provided evidence that the TRPV channel OSM-9 can be phosphorylated in vitro by the AKT-1 kinase (on OSM-T10), and that this T10 residue has a key function in vivo, in potentiating the micromolar H2O2-induced sensory response in ASH neurons. It is tempting to speculate that a similar mechanism could take place in PHA neurons to promote their sensitivity to micromolar H2O2, but we have not tested it. As suggested by R2, this has been speculated in the discussion (p. 10-11). 

Interestingly, while we were working on the revisions, the same group published an article showing that DLK/p38MAPK signaling regulates LITE-1 photoreceptor’s stability in ASH photosensory neurons (Zhang et al., 2022). Therefore, the involvement of PMK-1 in LITE-1 recycling in PHA neurons, as PMK-3 in ASH neurons, is another possibility. This has also been mentioned in the discussion (p. 11). All these questions should be addressed in the future, including the SKN-1 possible role proposed by R2.

---

## [Editor Report · Decision Letter 1]

24 Aug 2022

Distinct mechanisms underlie H2O2 sensing in C. elegans head and tail

PONE-D-22-08767R1

Dear Dr. QUINTIN,

We’re pleased to inform you that your manuscript has been judged scientifically suitable for publication and will be formally accepted for publication once it meets all outstanding technical requirements.

Kind regards,

Elise A. Kikis

Academic Editor

PLOS ONE

Additional Editor Comments (optional):

The authors have adequately addressed the reviewers comments and concerns, resulting in a stronger and study and clearer manuscript.
---

## [Editor Report · Acceptance letter]

6 Sep 2022

PONE-D-22-08767R1 

Distinct mechanisms underlie H2O2 sensing in *C. elegans* head and tail 

Dear Dr. Quintin:

I'm pleased to inform you that your manuscript has been deemed suitable for publication in PLOS ONE. Congratulations! Your manuscript is now with our production department. 

Kind regards, 

on behalf of

Dr. Elise A. Kikis 

Academic Editor

PLOS ONE